# Disrupting CD22-*cis*-ligand interactions ameliorates type 1 diabetes and graft rejection by expanding regulatory B cells

Wang Long[1,2], Ayaka Endo[2], Hashadi Nadeesha Walakulu Gamage[1,2,3], Tianyi Yang[2], Toru Suzuki[4,5], Takeshi Nitta[6], Hiromu Takematsu[7], Kenya Honda[8,9], Masatake Asano[1†], Takeshi Tsubata[1,2]*

1 Department of Pathology, Nihon University School of Dentistry, Chiyoda-ku, Tokyo, Japan, 2 Department of Immunology, Medical Research Laboratory, Institute of Integrated Research, Institute of Science Tokyo, Bunkyo-ku, Tokyo, Japan, 3 Department of Structural Biology, Medical Research Laboratory, Institute of Integrated Research, Institute of Science Tokyo, Bunkyo-ku, Tokyo, Japan, 4 Laboratory of Genome Editing for Biomedical Research, Medical Research Laboratory, Institute of Science Tokyo, Bunkyo-ku, Tokyo, Japan, 5 Animal Research Facilities, Bioscience Center, Institute of Science Tokyo, Bunkyo-ku, Tokyo, Japan, 6 Division of Molecular Pathology, Research Institute for Biomedical Sciences, Tokyo University of Science, Noda, Chiba, Japan, 7 Department of Molecular Cell Biology, School of Medical Sciences, Fujita Health University, Toyoake, Aichi, Japan, 8 Department of Microbiology and Immunology, Keio University School of Medicine, Shinjuku-ku, Tokyo, Japan, 9 RIKEN Center for Integrative Medical Sciences, Yokohama, Kanagawa, Japan

† Deceased.
* tsubata.imm@mri.tmd.ac.jp

## Abstract

CD22 is an inhibitory receptor expressed in B cells and is constitutively associated with α2,6-sialylated membrane proteins expressed on the same cell (*cis*-ligands). However, interaction with *cis*-ligands is required for the function of CD22 only in part. To address the role of ligand interaction of CD22 in immune responses, here we generated anti-CD22 antibody 1C5 that specifically inhibits ligand binding of CD22. Both $Cd22^{-/-}$ mice and mice treated with 1C5 show expansion of regulatory B (Breg) cells in follicular (FO) B cells, suggesting a crucial role of ligand interaction of CD22 in inhibiting the expansion of FO Breg cells. CD22 appears to recognize BCR and TLRs thereby directly or indirectly suppressing TLR signaling essential for expansion of Breg cells. Treatment of mice with 1C5 ameliorates skin graft rejection and type 1 diabetes with expansion of regulatory γδ T cells probably through expansion of Breg cells, suggesting ligand interaction of CD22 as a novel target of therapy for autoimmune diseases and graft rejection.

## Introduction

CD22, a member of the sialic acid-binding immunoglobulin-like lectin (Siglec) family, is an inhibitory B cell receptor that negatively regulates B cell receptor (BCR) signaling [1–3]. In the absence of BCR ligation, BCR transmits a low-level signaling called

**Data availability statement:** All relevant data are within the paper and its Supporting information files. The numerical Data are provided in supporting S1 Data for each main figures and supplemental figures. All flow cytometry data files and corresponding gating strategy have been deposited and publicly available on Zenodo at https://zenodo.org/records/18490746.

**Funding:** This work was supported by Japan Society for the Promotion of Science (https://www.jsps.go.jp/english/) Grant-in-Aid for Scientific Research 26293062, 18H02610, 21K19373 and 21H02679 (TT). The funder had no role in study design, data collection and analysis, decision to publish, or preparation of the manuscript.

**Competing interests:** I have read the journal's policy and the authors of this manuscript have the following competing interests: TT is a member of PLOS Biology's Editorial Board. The other authors declare that no competing interests exist.

**Abbreviations**: BCR, B cell receptor; Breg cell, regulatory B cell; BSA, bovine serum albumin; CFSE, carboxyfluorescein diacetate succinimidyl ester; CyA, Cyclosporin A; DC, dendritic cell; FCS, fetal calf serum; FO, follicular; IFN-γ, interferon-γ; IL, interleukin; i.p., intraperitoneal; ITIMs, immunoreceptor tyrosine-based inhibition motifs; LPS, lipopolysaccharide; MDSC, myeloid-derived suppressor cell; MZ, marginal zone; OVA, ovalbumin; PBs, plasmablasts; PCs, plasma cells; PMA, Phorbol myristate acetate; SHP-1, SH2-containing protein tyrosine phosphatase 1; Siglec, sialic acid-binding immunoglobulin-like lectin; Tem, effector memory T; T1D, type 1 diabetes; TCR, T cell receptor; TGF-β, transforming growth factor-β; TLR, toll-like receptor; TNF-α, tumor necrosis factor-α.

tonic signaling that regulates B cell survival and differentiation [4]. CD22 inhibits tonic signaling as well as BCR ligation-induced signaling [5–7]. CD22 contains immunoreceptor tyrosine-based inhibition motifs (ITIMs) in the cytoplasmic region [1,8]. Upon phosphorylation, CD22 ITIMs recruit and activate SH2-containing protein tyrosine phosphatase 1 (SHP-1) [9], which negatively regulates BCR signaling by dephosphorylating signaling molecules such as Syk [10]. CD22 also inhibits B cell activation induced by various stimuli including toll-like receptor (TLR) ligands [9,11,12].

Ligand recognition of Siglecs is unique in that the extracellular lectic domain in Siglecs recognizes various sialylated ligands and is constitutively occupied by the ligands expressed on the same cell (*cis*-ligands) although Siglecs also recognize the ligands expressed in other cells (*trans*-ligands) [13]. CD22 recognizes α2,6 sialylated *cis*-ligands such as IgM, CD45, and CD22 [3]. *Cis*-ligand interaction of CD22 regulates the CD22 activity either positively or negatively depending on the *cis*-ligands. Homotypic CD22 interaction negatively regulates the CD22 activity [14,15], but this regulation requires strong CD22 phosphorylation by BCR ligation [15]. In contrast, ligand interaction of CD22 is involved in CD22-mediated inhibition of tonic BCR signaling [5]. This inhibition appears to be mediated by the recognition of BCR by CD22 as a ligand [16], leading to the association of these molecules to CD22. Ligand interaction of CD22 also inhibits B cell activation by TLR ligands [9,12]. Although TLRs are not associated with CD22, CD22 might also be a ligand of CD22 because TLRs are recognized by other Siglecs such as Siglec-10 and Siglec-E [17,18]. Previously, we showed that the reduction of tonic signaling by CD22 *cis*-ligands plays a role in the elimination of signaling-incompetent B cells [5]. The tonic signaling level in signaling-incompetent cells may become insufficient for survival by the reduction of tonic signaling. However, the basal intracellular $Ca^{2+}$ level in $Cd22^{-/-}$ B cells is higher than that in $St6gal1^{-/-}$ B cells that lack α2,6 sialic acid [5], suggesting that tonic signaling in B cells is reduced by CD22 through ligand-dependent and independent pathways.

*Cd22*$^{-/-}$ mice show various phenotypes in immune responses including defective B cell response to T cell-independent antigens [19,20] and expansion of regulatory (Breg) cells [21]. Breg cells comprise a fraction of various mouse and human B cell subsets and characteristically produce the inhibitory cytokine interleukin-10 (IL-10) [22–25]. IL-10 plays a role in the immune inhibitory activity of Breg cells, but their inhibitory activity involves other mechanisms such as the production of the inhibitory cytokines IL-35 [26,27], and transforming growth factor-β (TGF-β) [28,29], expression of CD39 and CD73 that generate the immunosuppressive molecule adenosine [30–32], and cytotoxicity of T cells [33]. It is already established that Breg cells suppress various immune responses including autoimmunity, graft rejection, allergy, cancer immunity, and infection immunity [22–24]. Breg cells are shown to be expanded by toll-like receptor (TLR) ligands such as lipopolysaccharide (LPS) [21,34] and cytokines such as IL-6 [35,36] and IL-21 [37]. These TLR ligands and cytokines induce inflammation and therefore cannot be used for the treatment of autoimmune diseases or graft rejection. T-cell immunoglobulin and mucin domain 1 (Tim-1), a receptor for phosphatidyl serine, and SLAMF5 (CD84), a member of the SLAM family, IL-35 and thioredoxin are shown to regulate expansion of Breg cells. However, these molecules are known to regulate various immune cell types.

Because *Cd22*[−/−] mice show expansion of Breg cells [21], CD22 appears to suppress the expansion of Breg cells thereby augmenting immune responses. However, whether the ligand interaction of CD22 is involved in the regulation of Breg cells is not yet clear because CD22 inhibits tonic signaling by ligand-independent as well as dependent pathways [5]. To address the role of ligand interaction of CD22 in the CD22-mediated regulation of immune responses, we generated an anti-mouse CD22 antibody that inhibits ligand binding of CD22. We show that treatment with this antibody augments tonic signaling and activation of B cells by a TLR ligand probably by inhibiting the CD22 activity. Treatment of mice with this anti-CD22 antibody increases the number of Breg cells, ameliorates type 1 diabetes (T1D) in a mouse model, and prolongs skin allograft survival. These results indicate that the ligand interaction of CD22 plays a crucial role in the regulation of autoimmune responses and graft rejection by limiting the expansion of Breg cells. Because CD22 is primarily expressed in B cells, ligand interaction of CD22 is a promising target of therapies for autoimmune diseases and graft rejection.

## Results

### In vitro treatment with anti-CD22 antibody 1C5 that inhibits ligand-binding of CD22 modulates the activity of CD22

We first generated anti-CD22 antibodies 1C5, 1D9, 1D7, and 2C9 by establishing hybridomas using *Cd22*[−/−] mice immunized with the mouse CD22-Fc protein (S1A Fig). The anti-CD22 1C5 but not the other anti-CD22 antibodies including previously established F239 [16] inhibited the binding of CD22 to the ligands (S1B and S1C Fig). Interaction of CD22 with *cis*-ligands was shown to be involved in the CD22-mediated inhibition of B cell activation induced by LPS and CD40 [12]. Indeed, B cell survival and proliferation induced by anti-CD40 or LPS is augmented in *Cd22*[−/−] B cells (Figs 1A, 1B, S2A, and S2B) and by treatment with the synthetic sialoside GSC718 that inhibits ligand binding of CD22 (Figs 1C, 1D, S2C, and S2D) as shown previously [9,11,12]. To address whether 1C5 regulates the function of CD22, we thus stimulated mouse B cells in vitro with anti-CD40 or LPS with or without anti-CD22 antibodies. B cell survival and proliferation induced by anti-CD40 or LPS is significantly augmented by treatment with 1C5 but not the isotype-matched control antibody C6 (Figs 1E, 1F, S2E, and S2F). None of the anti-CD22 that do not inhibit ligand binding augments B cell proliferation induced by LPS or anti-CD40. This result indicates that 1C5 but not other anti-CD22 efficiently regulates CD22-mediated suppression of B cell activation induced by LPS and CD40 signaling. In contrast, the interaction of CD22 with *cis*-ligands has been shown to up-regulate BCR ligation-induced signaling by mitigating CD22-mediated signal inhibition [12]. Treatment with 1C5 but not other anti-CD22 inhibits anti-IgM-induced B cell survival and proliferation as well as GSC718, suggesting that 1C5 regulates CD22-mediated suppression of BCR ligation-induced signaling by inhibiting the *cis*-ligand interaction of CD22. Taken together, 1C5 efficiently regulates the function of CD22 by inhibiting the *cis*-ligand binding of CD22.

### In vivo treatment with 1C5 modulates the CD22 activity

Next, we addressed whether the in vivo treatment of mice with 1C5 regulates the CD22 activity. Treatment with 1C5 does not reduce the percentage of B cells in the spleen (S3A Fig), indicating that 1C5 does not delete B cells probably because 1C5 is IgG1, which induces complement and Fc receptor-mediated cytotoxicity only weakly. *Cd22*[−/−] B cells show altered phenotypes including reduction of both surface IgM and IgD in follicular (FO) B cells (Fig 2A), expansion of CD5[+] B cells (Fig 2B), and reduction in marginal zone (MZ) B cells (S3B Fig) as shown previously [19,20,38,39]. These phenotypic changes are suggested to be induced by higher tonic signaling and are shown to depend on the ligand interaction of CD22 [5]. The surface IgM level in FO B cells is reduced by in vivo treatment with anti-CD22 regardless of ligand binding inhibition (Fig 2C). In contrast, the surface IgD level in FO B cells is reduced by 1C5 but not the other anti-CD22 antibodies (Fig 2C). The percentage of CD5[+] B cells is increased by treatment with 1C5 but not other anti-CD22 antibodies (Fig 2D). The percentage of MZ B cells in wild-type mice treated with 1C5 is similar to that in *Cd22*[−/−] mice (S3B and S3C Fig). MZ B cells are almost completely lost by treatment with anti-CD22 antibodies that do not inhibit ligand binding (S2C Fig), in agreement with the previous finding [40]. Thus, in vivo treatment with 1C5 induces various phenotypic changes found in *Cd22*[−/−] B cells whereas only some of them are induced by the other anti-CD22 antibodies. The surface expression

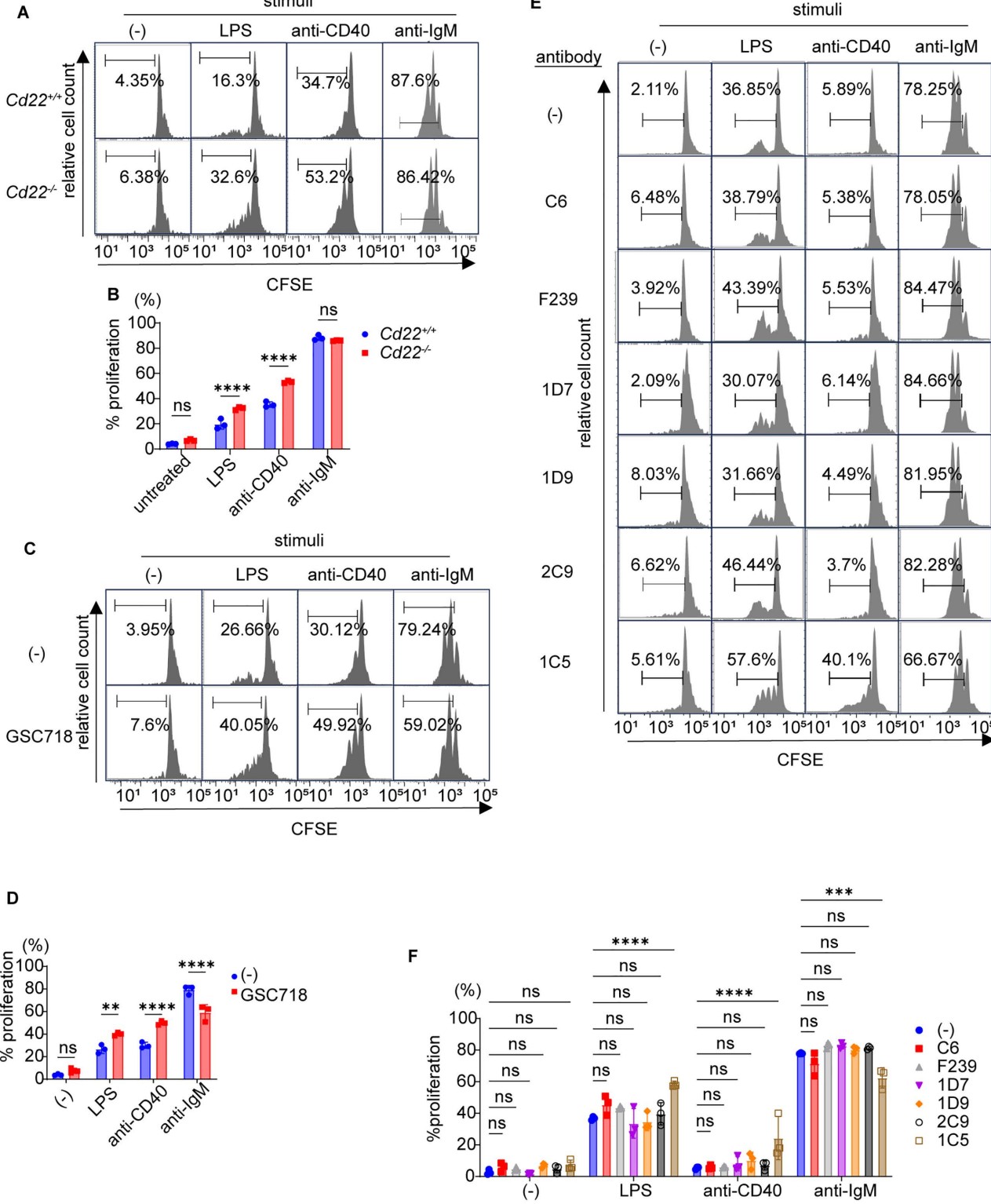

**Fig 1. 1C5 modulates B cell activation in vitro by inhibiting ligand binding of CD22.** Spleen B cells from 8 to 14 weeks old *Cd22*⁺/⁺ (**A–F**) or *Cd22*⁻/⁻ (A, B) mice were stained with 10 μM carboxyfluorescein diacetate succinimidyl ester (CFSE) and cultured with 10 μg/mL anti-mouse IgM antibody, 0.1 μg/mL LPS or 10 μg/mL anti-CD40 antibody for 72 hours in the presence or absence of 10 μM GSC718, 100 μg/mL anti-CD22 antibodies F239, 1D7,

1D9, 2C9, and 1C5 or an isotype-matched control antibody C6 for 72 hours. Cells were analyzed by FCM. Representative data are shown (A, C, E). Percentages of proliferated cells are indicated. Percentages of proliferated cells and Mean ± SD are shown ($n = 3$) (B, D, F). Data were analyzed by two-way ANOVA with Šidák (B, D) or Dunnett's (F) multiple comparison test. **$p < 0.01$, ***$p < 0.001$, ****$p < 0.0001$, ns, not significant. Data are representative of two (A–D) or three (E and F) independent experiments. Numerical data underlying this figure can be found in S1 Data, sheet "Fig 1".

of CD22 is reduced by both in vitro and in vivo treatment with all the anti-CD22 antibodies (S3D and S3E Fig). Because ligand interaction of CD22 up-regulates the surface expression of CD22 [5,41], loss of ligand interaction may be involved in the reduction of surface CD22 in 1C5-treated B cells. Ligation of CD22 by the other anti-CD22 may induce endocytosis of CD22. Nonetheless, the level of surface CD22 in 1C5-treated mice is comparable to those in mice treated with the other anti-CD22 (S3E Fig). Therefore, 1C5 inhibits the CD22-mediated regulation of B cell phenotypes more extensively than the other anti-CD22 antibodies, probably by inhibiting the ligand binding of CD22, although the reduction in surface CD22 may be involved in the restricted phenotypic changes of B cells induced by the other anti-CD22 antibodies.

## In vivo treatment with 1C5 expands follicular Breg cells

Because CD22 reduces Breg cells [21], we next addressed whether in vivo treatment with 1C5 expands Breg cells. We determined Breg cells capable of producing IL-10 using B cells expressing the IL-10 reporter $Il10^{Venus}$ gene after stimulation with phorbol myristate acetate (PMA)/ionomycin for five hours as described previously [21,36]. The percentage of IL-10-producing cells in spleen B cells and the number of IL-10+ spleen B cells are increased in $Cd22^{-/-}$ B cells compared to $Cd22^{+/+}$ B cells (Figs 3A and S4A) in agreement with the previous finding [21]. The percentage of IL-10+ cells in spleen B cells and the number of IL-10+ spleen B cells are increased by in vivo treatment with 1C5 but not other anti-CD22 antibodies (Figs 3B and S4B), suggesting that inhibition of ligand binding of CD22 expands Breg cells probably by strongly inhibiting the CD22 activity.

Because the expression of markers such as CD21, CD23, CD43, CD1d, IgM, IgD, and CD5 is not modulated by PMA/ionomycin treatment (S5A–S5E Fig), we addressed the B cell population responsible for the generation of Breg cells by 1C5. When we examined IL-10+ cells in CD21loCD23hi FO B cells and CD21hiCD23lo MZ B cells, the number and percentage of IL-10+ cells were increased in FO B cells but not MZ B cells in both $Cd22^{-/-}$ and 1C5-treated mice (Figs 3C–3H and S4C), thereby the majority of IL-10+ B cells are FO B cells in these mice. This is confirmed by the increase in IL-10+ cells in IgMlo and IgDhi spleen cells in both $Cd22^{-/-}$ and 1C5-treated mice (Fig 4A and 4B). In both mice, IL-10+ cells are accumulated in CD5+, CD1dlo, and CD43− B cells (Fig 4A and 4B). This phenotype is not consistent with the initial observation that IL-10+ cells were accumulated in CD5+CD1dhi population [21]. However, the presence of IL-10+CD5+CD1dlo B cells was demonstrated later [42]. Although CD5 is preferentially expressed in B-1 cells, IL-10+ cells expanded by CD22 deficiency or treatment with 1C5 appear to be FO B cells because of a lack of CD43, a hallmark of B-1 cells [43] and expression levels of IgM, IgD, CD21, and CD23. Presence of CD5+ FO B cells was confirmed by the presence of B220+CD21+CD23+CD5+ cells in $Cd22^{-/-}$ spleen cells although these cells are very few in $Cd22^{+/+}$ spleen cells (S6 Fig), indicating that CD5+ FO B cells emerge through up-regulation of CD5 expression by suppression of CD22 function. These results indicate that in vivo treatment with 1C5 expands Breg cells in a subpopulation of FO B cells that become CD5-positive. When we calculated the fraction of IL-10+ FO B cells in total IL10+ B cells, IL10+ FO B cells are the major IL10+ B cell population in both $Cd22^{-/-}$ and 1C5-treated $Cd22^{+/+}$ mice but not in untreated or C6-treated $Cd22^{+/+}$ mice (Fig 3F and 3H), indicating that follicular Breg cells are a major Breg population in mice treated with anti-CD22 that inhibits ligand binding.

Because Tim-1 is shown to be expressed in Breg cells [44], we addressed Tim-1 expression in $Cd22^{+/+}$ and $Cd22^{-/-}$ IL-10+ B cells. In these mice, IL-10+ cells are mostly Tim-1+, and the percentages of Tim-1+ IL-10+ cells, and Tim-1+CD5+ cells are selectively increased in $Cd22^{-/-}$ B cells (S7 Fig). This result indicates that CD22 deficiency expands Tim-1+IL-10+ B cells.

We next examined the expression of CD22 and CD5 in various spleen B cells subsets. CD21hiCD23+ marginal zone precursors show higher CD22 expression compared to the other subsets in consistent with a previous study [45] (S6A

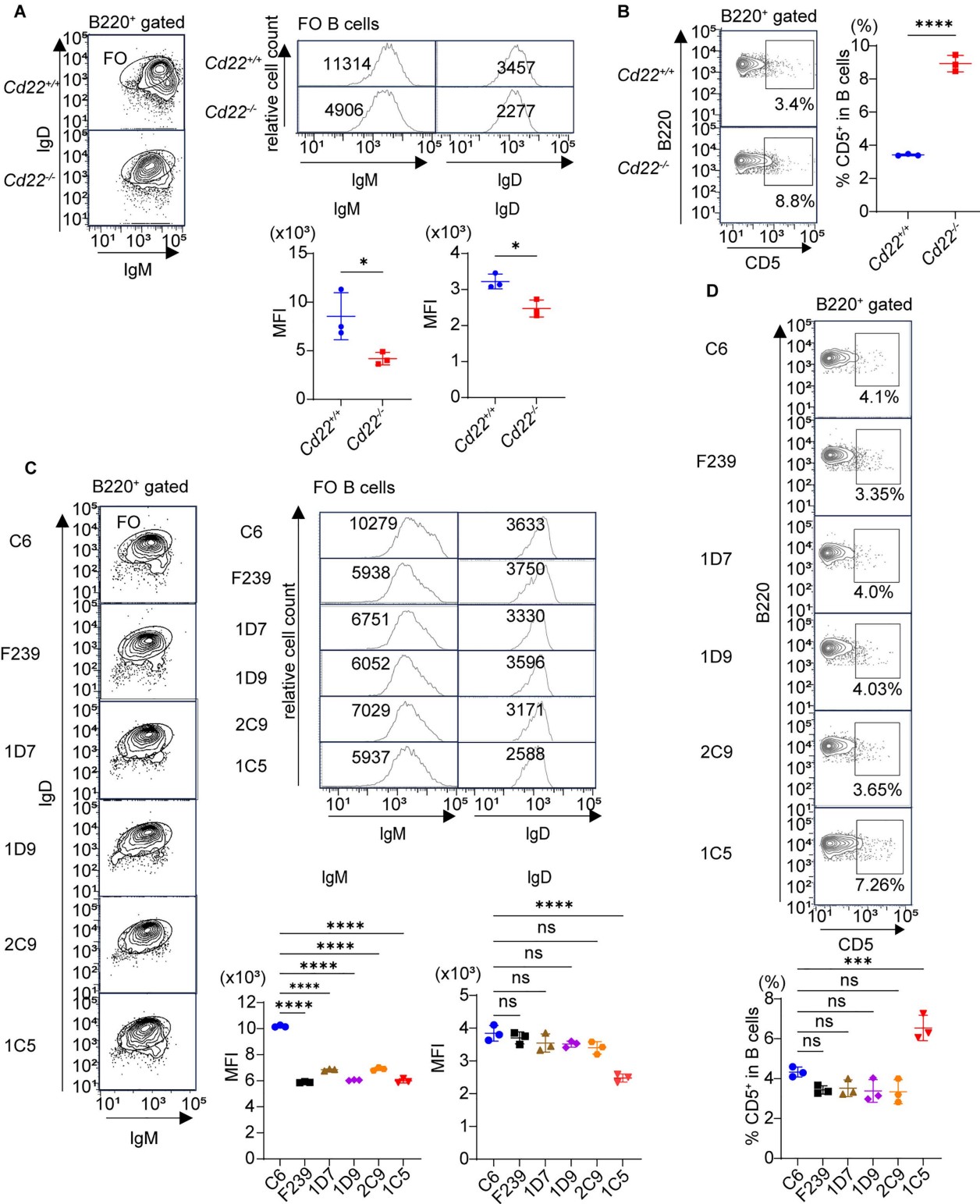

**Fig 2. Treatment with 1C5 inhibits the activity of CD22 in vivo.** Spleen cells from 8-14 weeks old *Cd22*$^{+/+}$ and *Cd22*$^{-/-}$ C57BL/6 mice (**A**, **B**) and C57BL/6 mice injected twice (8 days and 1 day before analysis) with anti-CD22 antibodies F239, 1D7, 1D9, 2C9, or 1C5 or the isotype-matched control antibody C6 (**C**, **D**). B220$^+$ cells were analyzed for IgM and IgD by FCM, and IgM$^{lo}$IgD$^{hi}$ FO B cells were gated and analyzed for the IgM and IgD levels

(A, C). Representative data are shown. MFIs of IgM and IgD are indicated. MFIs and Mean ± SD are shown ($n = 3$). B220$^+$ cells were analyzed for CD5 by FCM (B, D). Representative data are shown. Percentages of CD5$^+$ B cells in total B cells are indicated. Percentages of CD5$^+$ B cells and Mean ± SD are shown ($n = 3$). Data were analyzed by unpaired $t$ test (A, B) or one-way ANOVA with Tukey's multiple comparison (C, D). *$p < 0.05$, ***$p < 0.001$, ****$p < 0.0001$, ns, not significant. Data are representative of two (A and B) or four (C and D) independent experiments. Numerical data underlying this figure can be found in S1 Data, sheet "Fig 2".

Fig). In contrast, CD43$^+$CD23$^-$ B-1 cells show reduced CD22 expression. Thus, preferential induction of Breg cells in FO B cells by 1C5 may not be solely attributed to the level of CD22 expression.

We next addressed IL10$^+$ plasmablasts (PBs) and plasma cells (PCs) because these cells are known to be a major source of B cell-derived IL-10. In spleen cells from both $Cd22^{-/-}$ and 1C5-treated $Cd22^{+/+}$ mice, the percentages of B220$^{-/lo}$CD138$^+$ PCs and PBs are increased (S8 Fig). The percentage of PBs + PCs is not altered by treatment with PMA/ionomycin (S5F Fig), indicating that B220$^{-/lo}$CD138$^+$ cells can be analyzed as PBs/PCs even after treatment with PMA/ionomycin. In both $Cd22^{-/-}$ and 1C5-treated $Cd22^{+/+}$ mice, the percentage of IL10$^+$ PBs/PCs is significantly higher than those in untreated and C6-treated $Cd22^{+/+}$ mice (Fig 4C and 4D). These results indicate that 1C5 increases the total number of IL10$^+$ PBs/PCs by increasing the number of PBs/PCs and the percentage of IL-10$^+$ cells in PBs/PCs through CD22 inhibition.

## Treatment with 1C5 prevents the development of type 1 diabetes and controls insulitis

T1D and its mouse model in NOD mice are associated with defects in Breg cells [46]. We, therefore, addressed whether Breg expansion by 1C5 ameliorates T1D in NOD.CD72$^b$ mice that develop insulitis and diabetes with higher incidence and severity compared to NOD mice [47]. Because almost all NOD.CD72$^b$ mice develop insulitis at 12 weeks of age, we weekly injected 1C5 or the control antibody C6 from 6 to 13 weeks of age (Fig 5A). Histological analysis at 14 weeks of age shows much milder insulitis in 1C5-treated mice than the C6-treated mice (Fig 5B and 5C). Almost all C6-treated mice developed diabetes (Fig 5D) and died by 30 weeks of age (Fig 5E). In contrast, the blood glucose level is not increased in 1C5-treated mice although 1C5 is not injected after 14 weeks of age (Fig 5D). These results clearly show that 1C5 induces long-lasting tolerance and prevents the development of T1D.

To address whether 1C5 is effective on already developed insulitis, we weekly treated NOD.CD72$^b$ mice with 1C5 or C6 from 12 to 17 weeks of age (Fig 5F). 1C5-treated mice at 24 weeks of age show insulitis comparable to untreated mice at 12 weeks of age (Fig 5G and 5H) and do not show hyperglycemia (Fig 5I). In contrast, C6-treated mice at 24 weeks of age show much more severe insulitis (Fig 5G and 5H) than 1C5-treated mice at 24 weeks of age and untreated mice at 12 weeks of age, and most of them show hyperglycemia (Fig 5I). This result indicates that treatment with 1C5 stops further development of T1D even if mice are treated after insulitis is developed.

To address how 1C5 maintains self-tolerance in NOD.CD72$^b$ mice, we analyzed immune cells in 1C5 and C6-treated mice (Fig 6A). Total cell numbers (S9A Fig), as well as the percentages and the absolute numbers of B cells, T cells, CD4$^+$ T cells, CD8$^+$ T cells (Figs 6B, 6C, S10, and S11A), Treg cells (S9B, S10, and S11A Figs) and DCs (S9C, S10, and S11A Figs) in spleen and pancreatic lymph nodes are not different between C6 and 1C5-treated mice. We analyzed IL-10 production in unstimulated B cells using anti-IL-10 antibody although the detection efficacy is less sensitive than in B cells from $Il10^{Venus}$ reporter mice (S12 Fig). The percentage of IL-10$^+$ spleen B cells in 1C5-treated NOD.CD72$^b$ mice is higher than that in spleen B cells from untreated C57BL/6 mice but comparable to that in C6-treated NOD.CD72$^b$ mice (Figs 6E and S12). In C6-treated mice, Breg cells may be expanded due to inflammation.

The number of mononuclear cells isolated from the pancreas is lower in 1C5-treated mice compared to C6-treated mice (S9A Fig), consistent with milder insulitis in 1C5-treated mice (Fig 5B and 5C). In pancreas cells from 1C5-treated mice, the percentages of CD4$^+$ T cells, CD8$^+$ T cells (Fig 6C and 6D), Treg cells (S9B Fig) and DC (S9C Fig) are comparable to those from C6-treated mice whereas the numbers of these immune cells were lower than those from C6-treated

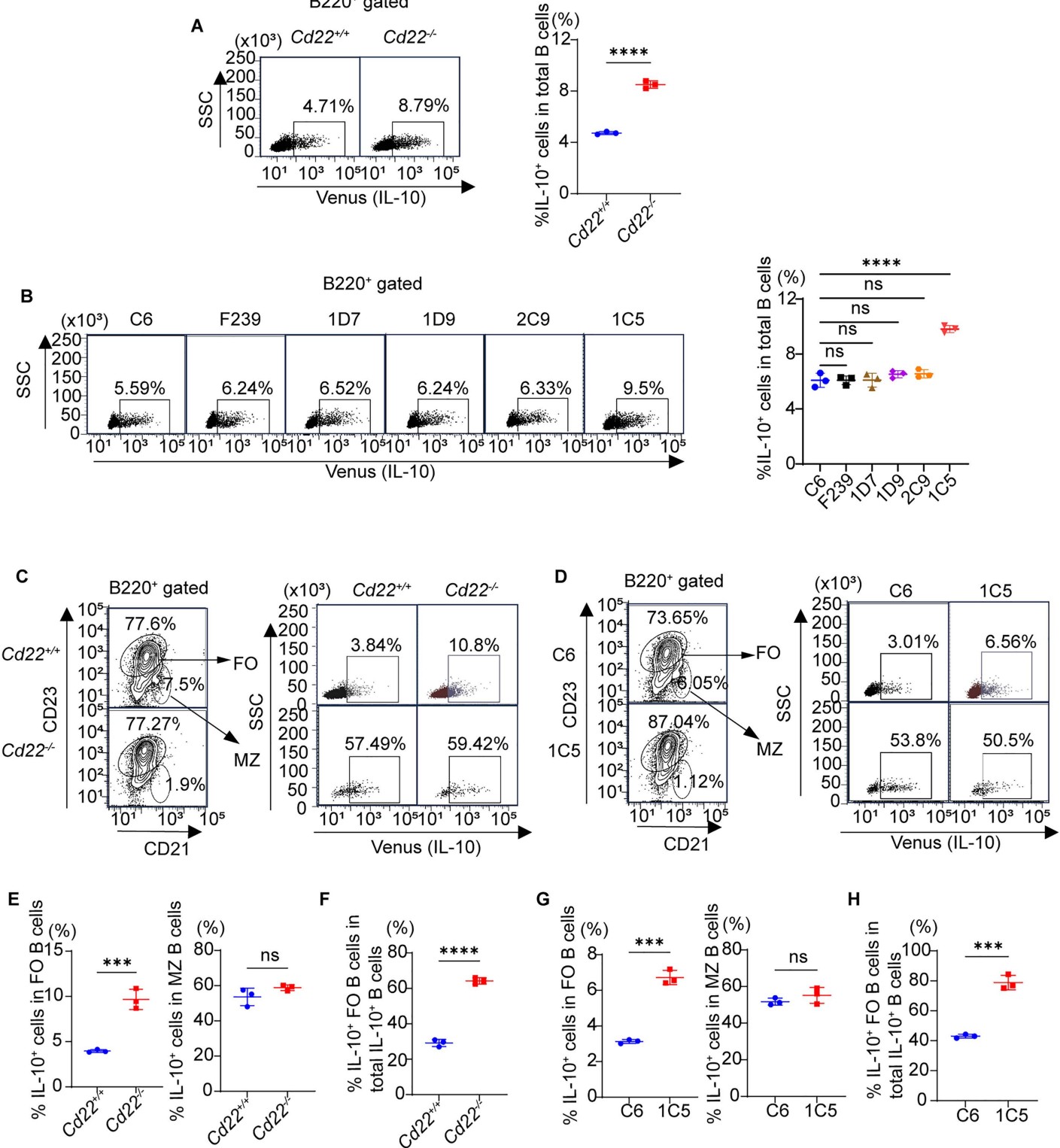

**Fig 3. In vivo treatment with 1C5 expands IL-10⁺ FO B cells.** Spleen cells from 8 to 14 weeks old *Cd22*⁺/⁺ **(A–H)** and *Cd22*⁻/⁻ (A, C, E, G) *Il10*^Venus^ reporter mice either untreated or injected twice (8 days and 1 day before analysis) with the anti-CD22 antibodies 1C5 (B, D, F, H), F239, 1D7, 1D9 or 2C9 (B) or the isotype-matched control antibody C6 (B, D, F, H) were stimulated with 10 ng/mL PMA and 1 μg/mL ionomycin for 5 hours. Total B220⁺

cells (A, B), CD21$^{int}$**CD23**$^{hi}$ FO B cells (C–H), and CD21$^{hi}$**CD23**$^{lo}$ MZ B cells (C-F) were analyzed for IL-10 by FCM. Representative data are shown (A–D). Percentages of IL-10$^+$ B cells and Mean ± SD are shown ($n$ = 3) (A, B, E–H). Data were analyzed by unpaired $t$ test (A, E–H) or one-way ANOVA with Tukey's multiple comparisons (B). ***$p$ < 0.001, ****$p$ < 0.0001, ns, not significant. Data are representative of two independent experiments (A–H). Littermate fluorescence controls for B cells from *Il10*$^{Venus}$ reporter mice are shown in **S12 Fig**. Numerical data underlying this figure can be found in **S1 Data**, sheet "Fig 3".

mice (S11B Fig) likely because of reduced cellularity. Although the percentage and the number of B cells in pancreases are reduced in 1C5-treated mice, the percentage of IL-10$^+$ pancreas B cells in 1C5-treated mice is higher than that in C6-Treated mice (Fig 6E and 6F). The number of IL-10$^+$ pancreas B cells is not higher (S11B Fig) because of lower cellularity. Nonetheless, the ratio of IL-10$^+$ Breg cells to CD62L$^-$CD44$^{hi}$ CD4$^+$ effector memory T (Tem) cells is roughly 5–1 and 1–10 in 1C5 and C6-treated pancreas, respectively, suggesting that Breg cells outnumber CD4$^+$ Tem cells, a potential target of Breg cells, through treatment with 1C5. The percentage and the number of CD4$^+$ Tem cells in C6-treated mice are extremely high, consistent with severe insulitis (Figs 6G, 6H, and S11B). In contrast, the percentage and the number of CD4$^+$ Tem cells are markedly lower in 1C5-treated mice (Figs 6G, 6H, and S11B). Interestingly, the percentage of CD3$^+$CD4$^-$CD8$^-$ cells is markedly increased in the pancreas in 1C5-treated mice (Fig 7A). These CD4$^-$CD8$^-$ T cells express γδ T cell receptor (TCR) (Fig 7A). The percentages and the number of γδ T cells are markedly increased in the pancreas from 1C5-treated mice compared to C6-treated mice (Figs 7A and S11B). Among Vγ1–7, most of these γδ T cells express Vγ6, and some of them express Vγ1 (S9D Fig). Upon stimulation with PMA/ionomycin, some of these γδ T cells produce IL-17 (Fig 7B). However, almost all of them express CD39 and most of them express TGF-β (Fig 7B). Although only a small fraction of the γδ T cells produce IL-10 (Fig 7B), expression of the immune suppressive molecules CD39 and TGF-β suggests a regulatory role of the expanded γδ T cells. However, percentages of T cells that produce interferon-γ (IFN-γ) and tumor necrosis factor-α (TNF-α) upon stimulation with PMA/ionomycin are not reduced in the spleen, pancreatic lymph nodes, and pancreas by 1C5 (S13 Fig). Because the number of immune cells in pancreases in 1C5-treated mice is much lower than that in C6-treated mice, the numbers of T cells capable of producing IFN-γ and TNF-α are reduced by 1C5. These results suggest that regulatory γδ T cells are induced by Breg cells and suppress the expansion and activation of immune cells in the pancreas thereby contributing to the inhibition of diabetes development in NOD.CD72$^b$ mice.

To address the role of the inhibition of ligand binding of CD22 in 1C5-induced alterations in immune cells in NOD.CD72$^b$ spleen, we treated these mice with anti-CD22 IgG1 1C5 or 2C9, the latter of which does not inhibit ligand binding of CD22 (S1 Fig). After weekly injection of these antibodies for 8 weeks (Fig 8A), mice treated with 1C5 but not 2C9 show marked reduction in the percentage of B cells (Fig 8B–8D), increase in the percentage of IL-10$^+$ B cells (Fig 8E and 8F), reduction in the percentage of CD4$^+$ Tem cells and increase in the percentage of naïve CD4$^+$ T cells (Fig 8G and 8H). Mice treated with 2C9 show markedly lower percentage of γδ T cells compared to 1C5-treated mice (Fig 9). These results clearly indicate that suppression of T cell activation and expansion of regulatory cells in pancreas in NOD mice requires inhibition of ligand binding of CD22.

## In vivo treatment with 1C5 elongates skin allograft survival

Because Breg cells are shown to suppress graft rejection [48], we addressed whether treatment with 1C5 elongates graft survival using skin allograft transplantation. We injected C57BL/6 mice with 1C5 or C6 once a week altogether four times, and then transplanted skin allograft from BALB/c mice. As a positive control, we injected the immunosuppressant cyclosporin A (CyA) in transplanted mice (Fig 10A). Treatment with 1C5 significantly delays the production of anti-allograft antibodies (Fig 10B) and elongates the survival of the graft to an extent similar to the treatment with CyA (Fig 10C and 10D), indicating that 1C5 suppresses graft rejection as strongly as CyA. Treatment with 1C5 suppresses atrophy of the graft (Fig 10E and 10F). However, there is no synergy between 1C5 and CyA (Fig 10B and 10C). When we used *Cd19*$^{Cre/+}$ *Il10*$^{flox/flox}$ mice in which IL-10 is specifically deficient in B cells as recipients (Fig 10A), 1C5 no longer elongates the survival of skin

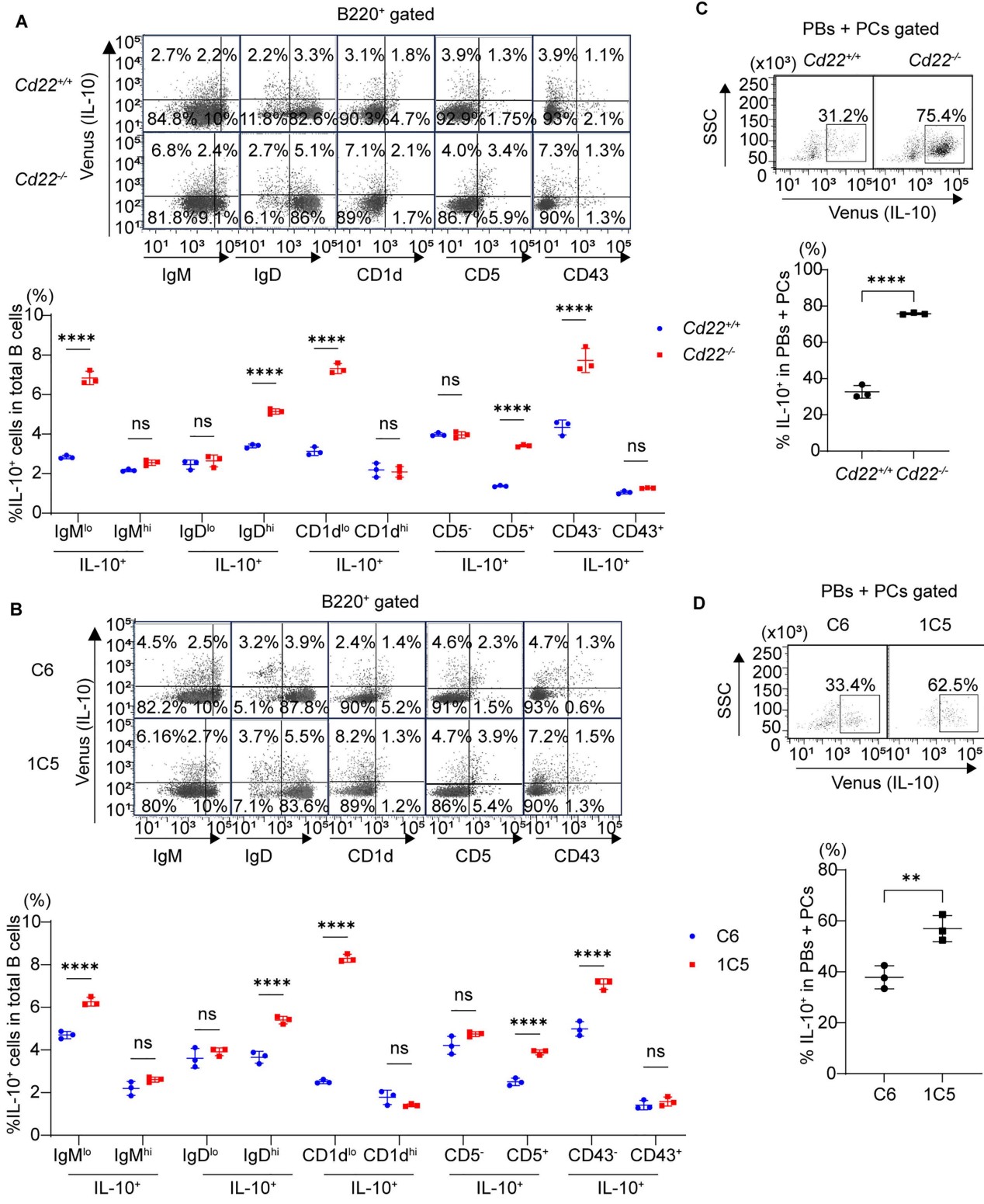

**Fig 4. IL-10 + B cells expanded by in vivo treatment with 1C5 show IgM<sup>lo</sup> IgD<sup>hi</sup> CD1d<sup>lo</sup> CD5+ CD43−.** Spleen cells obtained from 8-14 weeks old *Cd22<sup>+/+</sup>* (**A–D**) and *Cd22<sup>−/−</sup>* (**A, C**) *Il10<sup>Venus</sup>* reporter mice either untreated (A, C) or injected twice (8 days and 1 day before analysis) with the anti-CD22 antibody 1C5 or the isotype-matched control antibody C6 (B, D) were stimulated with 10 ng/mL PMA and 1 μg/mL ionomycin for 5 hours. B220<sup>+</sup> cells

were analyzed for IL-10, IgM, IgD, CD1d, CD5, and CD43 (A, B). Representative data are shown (upper panel). Percentages of each quadrant are indicated. Percentages of IL-10+ B cells positive or negative for the indicated markers in total B cells and Mean ± SD are shown (n = 3) (A, B, lower panels). Data were analyzed by one-way ANOVA followed by Tukey's multiple comparison test. Total B220−/loCD138+ PBs + PCs were analyzed for IL-10 by FCM (C, D). Representative data are shown (C, D, upper panels). Mean ± SD are shown (n = 3) (C, D, lower panels). Data were analyzed by unpaired t test. **p < 0.01, ****p < 0.0001, ns, not significant. Data are representative of two independent experiments (A–D). Numerical data underlying this figure can be found in S1 Data, sheet "Fig 4".

allograft (Fig 10G). This result indicates that IL-10 production from B cells is essential for the 1C5-induced suppression of graft rejection, and suggests that 1C5 elongates graft survival by expanding Breg cells. In both the draining lymph nodes and spleen, the total cell numbers (S14A Fig), as well as the percentages and the absolute numbers of B cells, CD4+ T cells, CD8+ T cells, Treg cells, DCs and γδ T cells are not altered in 1C5-treated recipients compared to C6-treated recipients (S14B, S14C, S15A, and S15B Figs). However, the percentage and number of IL-10+ B cells are increased in the draining lymph nodes but not the spleen in 1C5-treated recipients (Figs 10H and S15B), indicating local expansion of Breg cells. The percentages of T cells capable of producing TNF-α and IFN-γ are not altered (S14D and S14E Fig), but the percentage and the number of CD4+ Tem cells are reduced in the draining lymph nodes but not in spleen in 1C5-treated recipients (Figs 10I, S15A, and S15B). These results suggest that 1C5 locally expands Breg cells thereby suppressing graft rejection by inhibiting T cell activation.

### In vivo treatment with 1C5 does not alter antibody production to a T cell-dependent antigen

To address the role of CD22 cis-ligands in acute immune responses for foreign antigens, we treated C57BL/6 mice with 1C5 or C6 and immunized with the T cell-dependent antigen ovalbumin (OVA) (S16A Fig). The serum titer of anti-OVA IgG in 1C5-treated mice is comparable to that in C6-treated mice (S16B Fig), suggesting that 1C5 does not suppress immune responses to foreign antigens.

### Discussion

Interaction of CD22 with cis-ligands has been shown to either augment or suppress CD22-mediated signal inhibition depending on cis-ligands [14,15], and signal inhibition by CD22 involves both ligand-dependent and -independent pathways at least in the regulation of tonic signaling [5]. Here, we established the anti-CD22 antibody 1C5 that inhibits ligand binding of CD22. Mice treated with 1C5 show an increase in the number of B cells capable of producing IL-10 upon PMA/ionomycin treatment, strongly suggesting the expansion of Breg cells. In contrast, anti-CD22 antibodies that do not inhibit ligand binding of CD22 fail to expand Breg cells. These results suggest that the interaction of CD22 with endogenous ligands, probably cis-ligands, appears to suppress the expansion of Breg cells. Previously, CD22 was shown to suppress the expansion of Breg cells [21]. Therefore, the cis-ligand interaction of CD22 appears to play a crucial role in limiting the expansion of Breg cells by CD22. We further show that treatment with 1C5 ameliorates T1D in a mouse model and inhibits skin allograft rejection. 1C5-mediated allograft survival requires IL-10 production in B cells, strongly suggesting that Breg cells induced by 1C5 mediate graft survival. Taken together, the interaction of CD22 with endogenous ligands is essential for the CD22 function in suppressing the expansion of Breg cells thereby accelerating autoimmune responses and graft rejection.

Interaction of CD22 with endogenous ligands is required for CD22 to down-regulate both tonic signaling and TLR-mediated B cell activation [5,9,12]. CD22 recognizes IgM as a cis-ligand [15,16]. CD22 may also recognize TLRs as cis-ligands because other Siglecs such as Siglec-10 and Siglec-E recognize TLRs [17,18]. Recognition of IgM and TLRs by CD22 may induce association of these receptors with CD22 leading to CD22-mediated inhibition of tonic BCR signaling and TLR signaling. Because TLR signaling is crucial for the generation of Breg cells [21,34], the expansion of Breg cells may be suppressed by the interaction of CD22 with TLRs. Interestingly, we show that CD5+ Breg cells are expanded

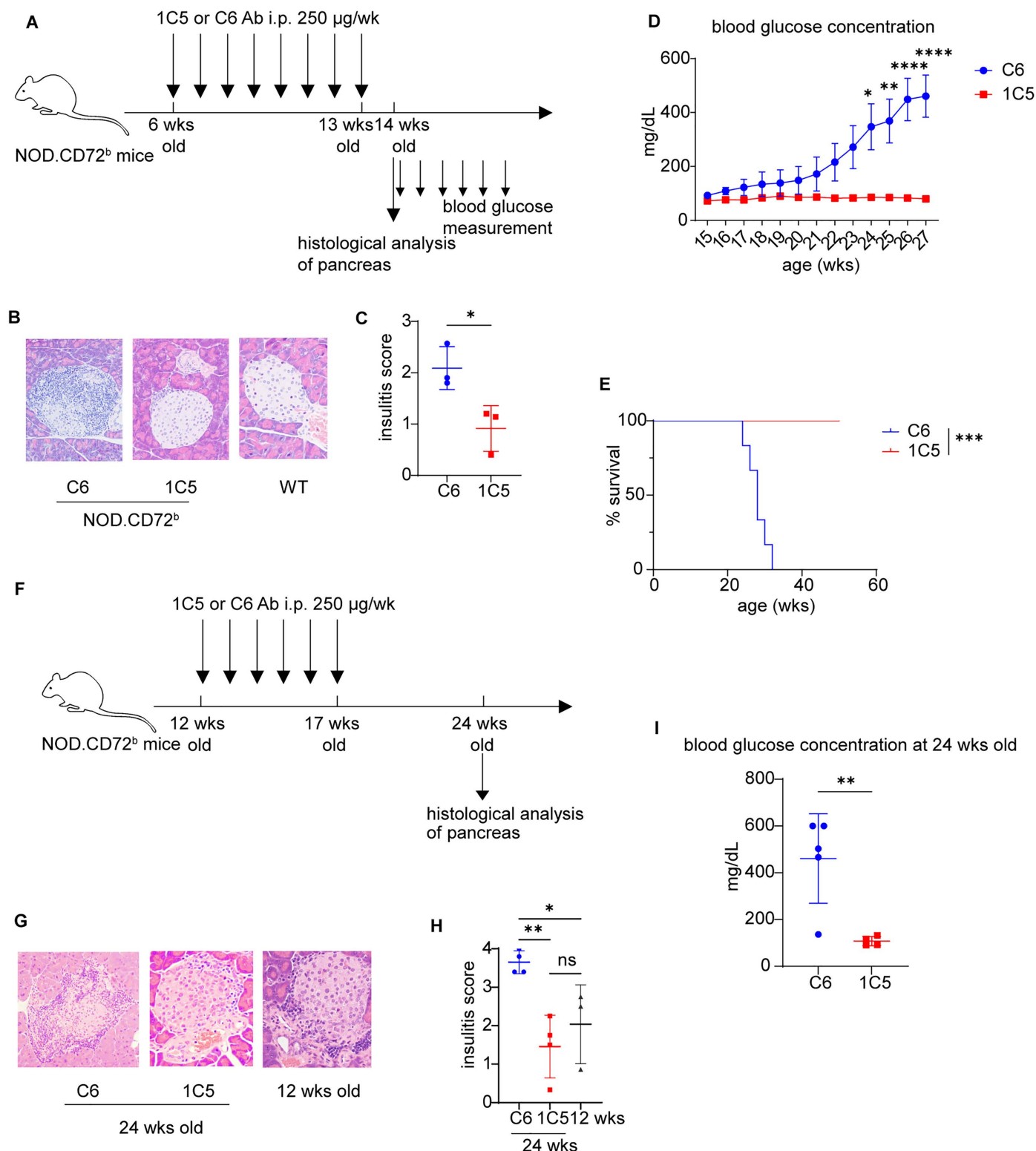

**Fig 5. In vivo treatment of NOD.CD72ᵇ mice with 1C5 inhibits the development of T1D. (A–E)** Prevention of insulitis and T1D by 1C5. A schematic representation of the experimental procedure is shown (A). Female NOD.CD72ᵇ mice were i.p. injected with 250 μg 1C5 or the control antibody C6

weekly from 6 to 13 weeks (wks) of age. Pancreas from antibody-treated NOD.CD72[b] mice and wild-type mice at 14 weeks of age were histologically analyzed (B). The severity of insulitis was scored as described in the Materials and methods section ($n = 3$). Insulitis score and Mean ± SD are shown ($n = 3$) (C). The blood glucose level was measured every week from 15 to 27 weeks of age (D). Mean ± SEM is shown ($n = 5$). The survival curve of 1C5 and C6-treated mice is shown ($n = 6$) (E), and data were analyzed by two-way ANOVA with Šidák multiple comparisons. (**F–I**) Treatment of female NOD. CD72[b] mice with 1C5 after insulitis development. A schematic representation of the experimental procedure is shown (F). Female NOD.CD72[b] mice were i.p. injected with 250 µg 1C5 or the control antibody C6 weekly from 12 to 17 weeks of age. Pancreas from antibody-treated or untreated NOD.CD72[b] mice at 24 weeks of age were histologically analyzed (G). The severity of insulitis was scored as described in the Materials and methods section (H). Insulitis score and Mean ± SD are shown ($n = 4$). Data were analyzed by one-way ANOVA followed by Tukey's multiple comparison test. The blood glucose level in antibody-treated mice was measured at 24 weeks of age (I). Mean ± SEM is shown ($n = 5$). Data were analyzed by unpaired $t$ test. $*p < 0.05$, $**p < 0.01$, $***p < 0.001$, $****p < 0.0001$. Data are representative of two independent experiments (A-I). Numerical data underlying this figure can be found in S1 Data, sheet "Fig 5".

in both $Cd22^{-/-}$ and 1C5-treated mice. Although CD5 is preferentially expressed in B-1 cells, these Breg cells appear to be FO B cells because they are CD21[lo]CD23[hi]IgM[lo]IgD[hi]CD43[−] (Fig 4A and 4B). These CD5[+] Bregs are likely generated from CD5[+] FO B cells through CD22 inhibition (S6 Fig). Previously, anergic B cells were shown to up-regulate the CD5 expression [49]. In these anergic B cells, CD5 expression may be induced by high steady-state signaling induced by the interaction of BCR with self-antigens. In $Cd22^{-/-}$ and 1C5-treated mice, tonic signaling increased by CD22 inhibition may generate CD5[+] FO B cells in which tonic signaling is particularly increased. BCR signaling is shown to crosstalk with TLR signaling [50]. Therefore, suppression of tonic signaling by the interaction of CD22 with BCR may down-regulate TLR signaling thereby indirectly inhibiting Breg expansion. Taken together, recognition of TLRs and BCR by CD22 as *cis*-ligands may directly or indirectly suppress TLR signaling thereby limiting the expansion of Breg cells.

Here, we show that in vivo treatment with 1C5 prevents the development of T1D in NOD.CD72[b] mice even when the treatment starts after insulitis development, and elongates skin allograft survival as effectively as CyA. In both pancreases in 1C5-treated NOD.CD72[b] mice and the draining lymph nodes in 1C5-treated skin-grafted mice, the percentage of CD4[+] Tem cells is markedly reduced, and the percentage of IL-10[+] B cells is increased compared to those in C6-treated mice. These results indicate that 1C5 expands Breg cells and inhibits the activation of T cells at the sites of the autoimmune response and the response to the allograft. Remarkably, 1C5-treated NOD.CD72[b] pancreas shows marked expansion of γδ T cells expressing CD39. CD39 is an ecto-enzyme involved in the generation of adenosine, which inhibits the activation of T cells and DCs [51]. Accumulating evidence suggests a regulatory role of CD39[+] γδ T cells. CD39[+] γδ T cells are shown to strongly inhibit T cell activation in vitro [52], and induce immune tolerance to the hepatitis B virus by expanding myeloid-derived suppressor cells and inducing CD8[+] T cell exhaustion [53]. γδ T cells secrete IL-17, and IL-17 is involved in their regulatory function [53]. Therefore, IL-17-secreting γδ T cells in 1C5-treated NOD.CD72[b] mice appear to have a regulatory activity. Because the number of regulatory γδ T cells far exceeds the number of Breg cells in pancreases in 1C5-treated NOD.CD72[b] mice, regulatory γδ T cells may play a central role in the suppression of insulitis in these mice. These phenotypic changes in IL-10[+] B cells, naïve versus effector CD4[+] T cells and γδ T cells are not induced by the treatment with the anti-CD22 antibody 2C9 that does not inhibit ligand binding of CD22, further supporting the essential role of the inhibition of ligand binding in the regulation of T1D by 1C5.

Because γδ T cells do not express CD22, 1C5 may not directly induce regulatory γδ T cells, but Breg cells expanded by 1C5 may be involved in the generation of regulatory γδ T cells. Breg cells suppress various immune cells but expand Treg cells by secreting IL-35 [26,27]. IL-35 is shown to induce exhaustion of γδ T cells [54]. However, it is not yet known whether IL-35 is involved in the generation of regulatory γδ T cells. Previously, TGF-β was shown to promote the generation of regulatory γδ T cells [55]. Because Breg cells secrete TGF-β as well as IL-10 and IL-35 [22,23], cytokines including TGF-β secreted from Breg cells may be involved in the expansion of regulatory γδ T cells. In allograft-transplanted mice, γδ T cells were not increased, suggesting that Breg-mediated immune suppression does not always associated with expansion of γδ T cells. How regulatory γδ T cells are generated by the inhibition of CD22 needs to be elucidated by future studies.

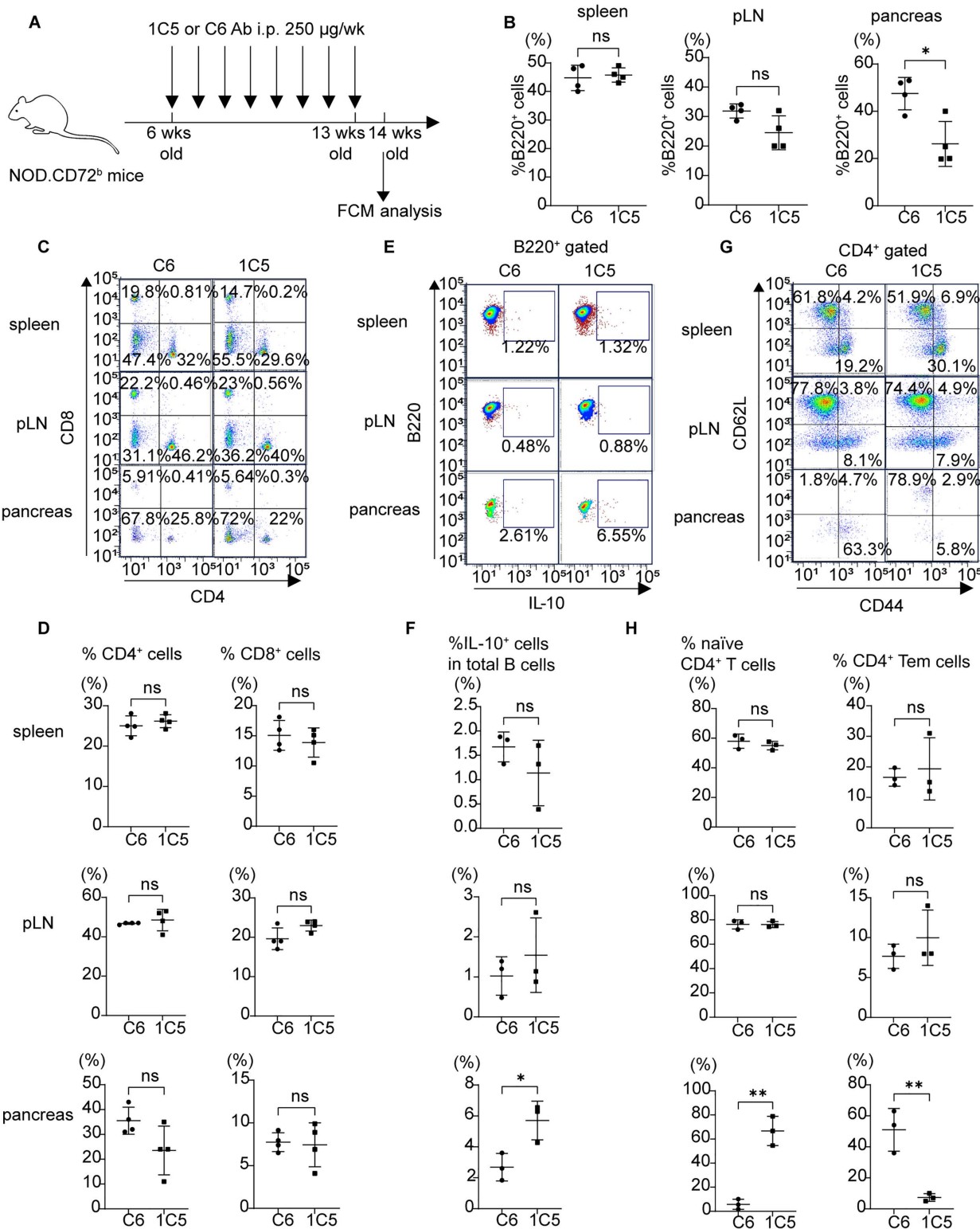

**Fig 6. In vivo treatment with1C5 expands Breg cells and reduces CD4 $^+$ Tem cells in the pancreas in NOD. CD72$^b$ mice. (A)** Schematic representation of experimental procedure. Female NOD.CD72$^b$ mice were i.p. injected with 250 μg 1C5 or the control antibody C6 weekly from 6 to14 weeks **(B-H)** FCM analysis for B220$^+$ B cells **(B)**, CD4$^+$CD8$^-$, CD4$^-$CD8$^+$ T cells **(C, D)**, IL-10$^+$ B220$^+$ B cells **(E, F)**, CD62L$^+$CD44$^{lo}$ naïve CD4$^+$ T cells, and

CD62L−CD44hi CD4+ Tem cells (G, H) in spleen, pancreatic lymph nodes and pancreas from 1C5 and C6-treated 15 weeks old mice. Representative data are shown. Percentages of CD4+, CD8+, CD4−CD8− T cells (D), IL-10+ B cells (F), CD62L+CD44lo naïve CD4+ T cells, and CD62L−CD44hi CD4+ Tem cells (H) are indicated. Mean±SD are shown (n=4) (B-H). Data were analyzed by two-way ANOVA with Šidák multiple comparison test. *$p<0.05$, **$p<0.01$, ns, not significant. Data are representative of two independent experiments (A-H). Numerical data underlying this figure can be found in S1 Data, sheet "Fig 6".

Once patients develop T1D, insulin secretion cannot be restored by suppression of insulitis [56]. As such, immune therapies to inhibit insulitis have not been developed as a treatment for T1D patients. Recently, insulitis has been detected at the prediabetic stage in those who have a risk of T1D, and clinical trials of immune therapy for these patients are undergoing [57]. Here, we show that treatment of mice with ligand-inhibiting anti-CD22 at the prediabetic stage of insulitis prevents the development of T1D. Cd22−/− mice show normal T cell-dependent antibody responses [19,20,38,39] and normal infection immunity to various pathogens [58,59], suggesting that CD22 inhibition may not cause general immune suppression. Indeed, 1C5 does not suppress T cell-dependent antibody response to immunized OVA. Immune suppression by 1C5 may require factors related to autoimmune responses and graft rejection, such as persistence of antigens. Moreover, it is unlikely that 1C5 regulates immune cell types other than B cells because CD22 is preferentially expressed in B cells. Therefore, ligand-inhibiting anti-CD22 antibodies may be an ideal therapy for the prediabetic stage of T1D to prevent the development of diabetes without impairing infection immunity. After organ transplantation, graft survival is induced by immunosuppressants such as CyA [60]. However, most of the recipients need continuous therapy with immunosuppressants, which suppress infection immunity [61]. Because 1C5 shows graft survival as effective as CyA, ligand-inhibiting anti-CD22 may induce graft survival as well as immunosuppressants without perturbing infection immunity. Therefore, ligand-inhibiting anti-CD22 has the potential to improve treatments of autoimmune diseases including T1D, and the prevention of graft rejection after organ transplantation.

## Materials and methods

### Ethics statement

All animal experiments were approved by the Institutional Animal Care and Use Committee, Institute of Science Tokyo (approval numbers: A2021-105 and A2023-215) or Animal Care and Use Committee, Nihon University (approval numbers: AP22DEN047-2, AP23DEN020-2 and AP22DEN031-2) and were performed according to the institutional guidelines.

### Mice

BALB/c and C57BL/6 mice were purchased from Sankyo Laboratory Service (Tokyo, Japan). Cd19cre/cre (a gift of Dr. Klaus Rajewsky) [62], Il10flox/flox [63] (a gift of Dr. Axel Roers), Il10Venus reporter [64], Cd22−/− [20] (a gift of Dr. Thomas F. Tedder), NOD.CD72b mice [47] were described previously. Another Cd22−/− mouse line was generated by CRISPR/Cas9-mediated genome editing using cloning-free CRISPR/Cas system as described previously [65]. For deleting the Cd22 exon 2–4, crRNAs were designed to target the introns between exons 1 and 2 (5′-GGAAGAAUUAGUCUGUAAACguuuuagagc ua ugcuguuuug-3′) and between exons 4 and 5 (5′-GAUGGAUUGAUUGAUGCUGGguuuuagagcuaugcuguuuug-3′). All the CRISPR/Cas9 components including crRNAs and a tracrRNA were injected into fertilized eggs obtained from C57BL/6 mice. Mice were maintained in the animal facility of Institute of Science Tokyo and Nihon University under specific pathogen-free conditions. Mice were intraperitoneally injected with 250 µg anti-CD22 antibodies (mouse IgG1) or isotype-matched control antibody C6 [66] (a gift of Dr. Takachika Azuma, Tokyo University of Science) every week. The concentration of blood glucose was measured by a glucometer (Accu-Chek Aviva).

### Cell culture

Mouse spleen B cells were prepared as described previously [67]. Cells were cultured in RPMI-1640 medium supplemented with 10% fetal calf serum (Nichirei Biosciences) 50 µM 2-mercaptoethanol (Sigma), 2 mM L-Glutamine (Wako),

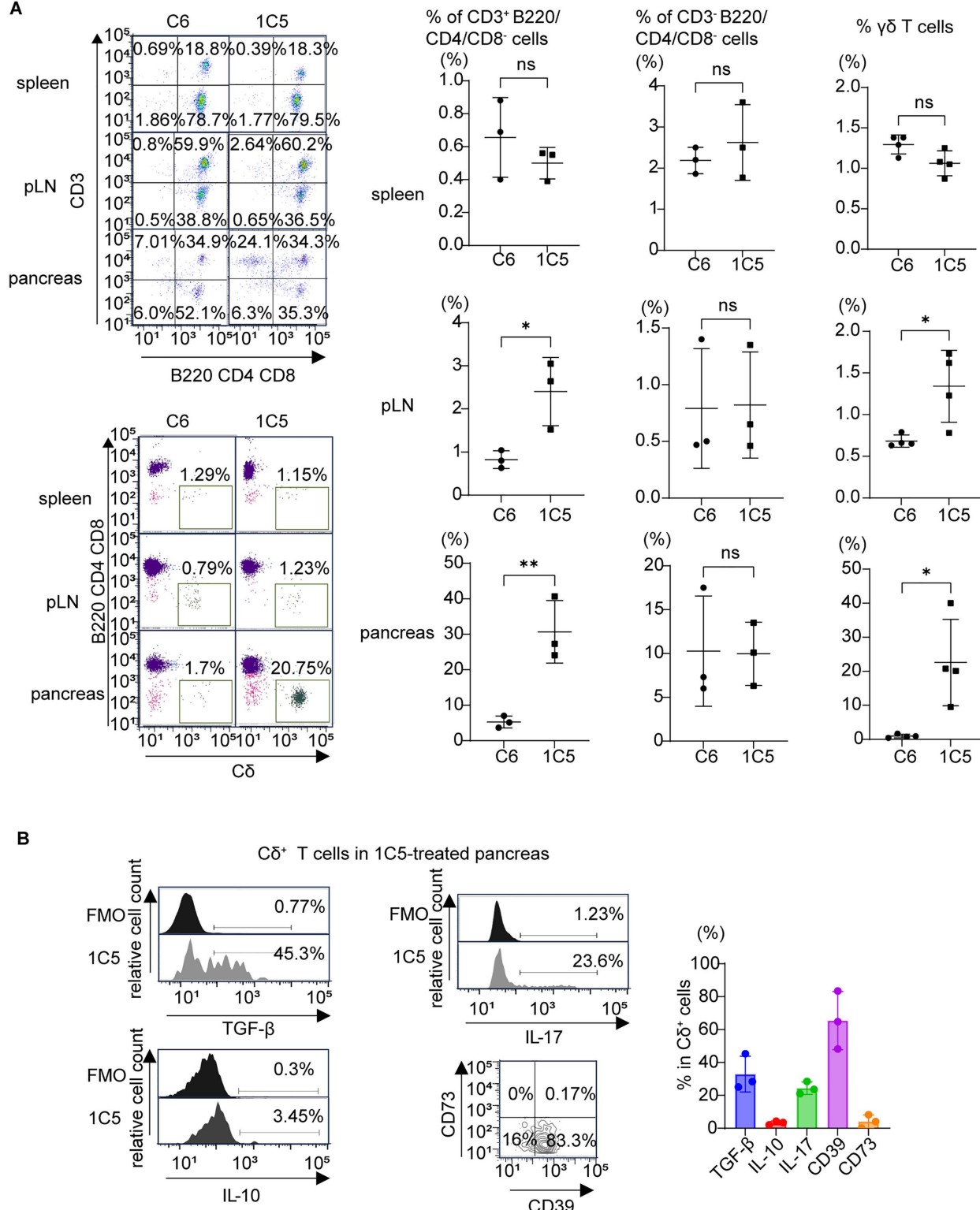

**Fig 7. In vivo treatment with 1C5 generates regulatory γδ T cells in the pancreas in NOD.** CD72[b] mice. Female NOD.CD72[b] mice were i.p. injected with 250 mg 1C5 or the control antibody C6 weekly from 6 to 14 weeks of age. **(A)** FCM analysis for Cδ⁺ γδ T cells in spleen, pancreatic lymph nodes, and pancreas from 1C5 and C6-treated 15-week-old mice. Percentages of γδ T cells are indicated. Mean ± SD are shown (*n* = 4). Data were analyzed by

two-way ANOVA with Šidák multiple comparison test. *$p < 0.05$, **$p < 0.01$, ns, not significant. Data are representative of two independent experiments. **(B)** FCM analysis for TGF-β, IL-10, IL-17, and CD39/CD73 in pancreas cells from 1C5-treated 15-week-old mice. Pancreas cells were in vitro stimulated with 10 ng/mL PMA and 1 µg/mL ionomycin for 5 hours and analyzed for TGF-β, IL-10, IL-17. CD39 and CD73 were analyzed in unstimulated pancreas cells. Representative data and Mean ± SD are shown (*$n = 3$*). Numerical data underlying this figure can be found in S1 Data, sheet "Fig 7".

and 1% penicillin/streptomycin (Nacalai). For induction of cytokine production, cells were cultured in the presence of 10 ng/mL PMA (Sigma) and 1 µg/mL ionomycin (Sigma) for 5 hours.

## Establishment of anti-CD22 hybridomas

The mCD22-Fc protein containing N-terminal 3 immunoglobulin domains of mouse CD22 and the Fc region of human IgG1 was described previously [68]. *Cd22*$^{-/-}$ mice were intraperitoneally immunized with mCD22-Fc together with complete Freund's adjuvant (Sigma). Mice were boosted with mCD22-Fc together with incomplete Freund's adjuvant (Sigma) at 2 weeks and 5 weeks after the primary immunization. Spleen cells from the immunized mice were fused with the mouse myeloma line PAI (a gift of Noriko Sorimachi) [69] using PEG 1500 (Roche) according to the standard protocol [70].

## Flow cytometry

Anti-mouse B220 (RA3-6B2) [71], anti-mouse CD8 (TIB-105) [72] anti-mouse Gr1 (RB6-8C5) [73], and anti-mouse CD22 (F239) [16] antibodies were conjugated with Alexa Fluor-647, Alex Fluor-488, and Pacific Blue using Antibody Labeling Kits (Invitrogen). Anti-mouse B220 (RA3-6B2) and anti-mouse CD8 (TIB-105) were biotinylated using EZ-Link NHS-LC-Biotin (Thermos Scientific). Pancreatic lymph nodes were isolated under the stereo microscope (Leica). Pancreas were minced and treated with 1 mg/mL Collagenase VI (Worthington Biochemical), and 1 ng/mL DNase I (Roche) in Hank's balanced salt solution (Thermo Fisher) at 37 °C for 20 min. Cells were passed through a 70-µm strainer (Funakoshi). In some experiments, after cell-surface staining, cells were fixed using 4% paraformaldehyde (TAAB) for 15 min, and permeabilized using PBS containing 0.2% saponin (Nacalai Tesque) and 3% bovine serum albumin (BSA) (Sigma) for 1h. For the detection of Foxp3, Foxp3/transcription Factor Staining Buffer Set (Invitrogen) was used for fixation and permeabilization. Cells were incubated with anti-FcγRII/III Ab 2.4G2 for 30 min to block FcγRII/III-mediated binding, and stained with the antibodies (S1 Table). Cells were analyzed by flow cytometry using FACS Verse (BD), and data were analyzed with FlowJo software (Tree Star).

## Cell proliferation assay

Cells were labeled with 10 µM CFSE (molecular probes) for 10 min. CFSE-labeled were cultured in 200 µL complete RPMI 1640 medium in 96-well plate with 10 µg/mL F(ab')$_2$ fragments of goat anti-mouse IgM antibody (Jackson ImmunoResearch), 0.1 µg/mL LPS (Sigma, E. *coli*, O111:B4) or 10 µg/mL anti-CD40 antibody (FGK45) [74] (a kind gift of Dr. Rolink), in the presence or absence of anti-CD22 antibodies for 72 h. The percentage of cells with reduced CFSE fluorescence was measured as divided cells.

## CD22 ligand binding inhibition assay

GSC718 was described previously [75]. mCD22-Fc in PBS (1 µg/mL) was incubated with 1 µg/mL anti-CD22 antibodies or GSC718 (final 1 µM) for 2 hours. The mixture was incubated with purified spleen B cells from C57BL/6 mice for 2 hours. Cells were stained with PE-goat anti-human IgG (SBA) at 4°C for 1 h and analyzed by flow cytometry using FACS Verse (BD).

## OVA Immunization

C57BL/6 mice were intraperitoneally injected with 250 µg 1 mg/mL C6 or 1C5 in PBS at 8 days and 1 day before immunization. Mice were subcutaneously immunized with 10 µg OVA (Sigma) in 100 µL PBS mixed with 100 µL alhydrogel

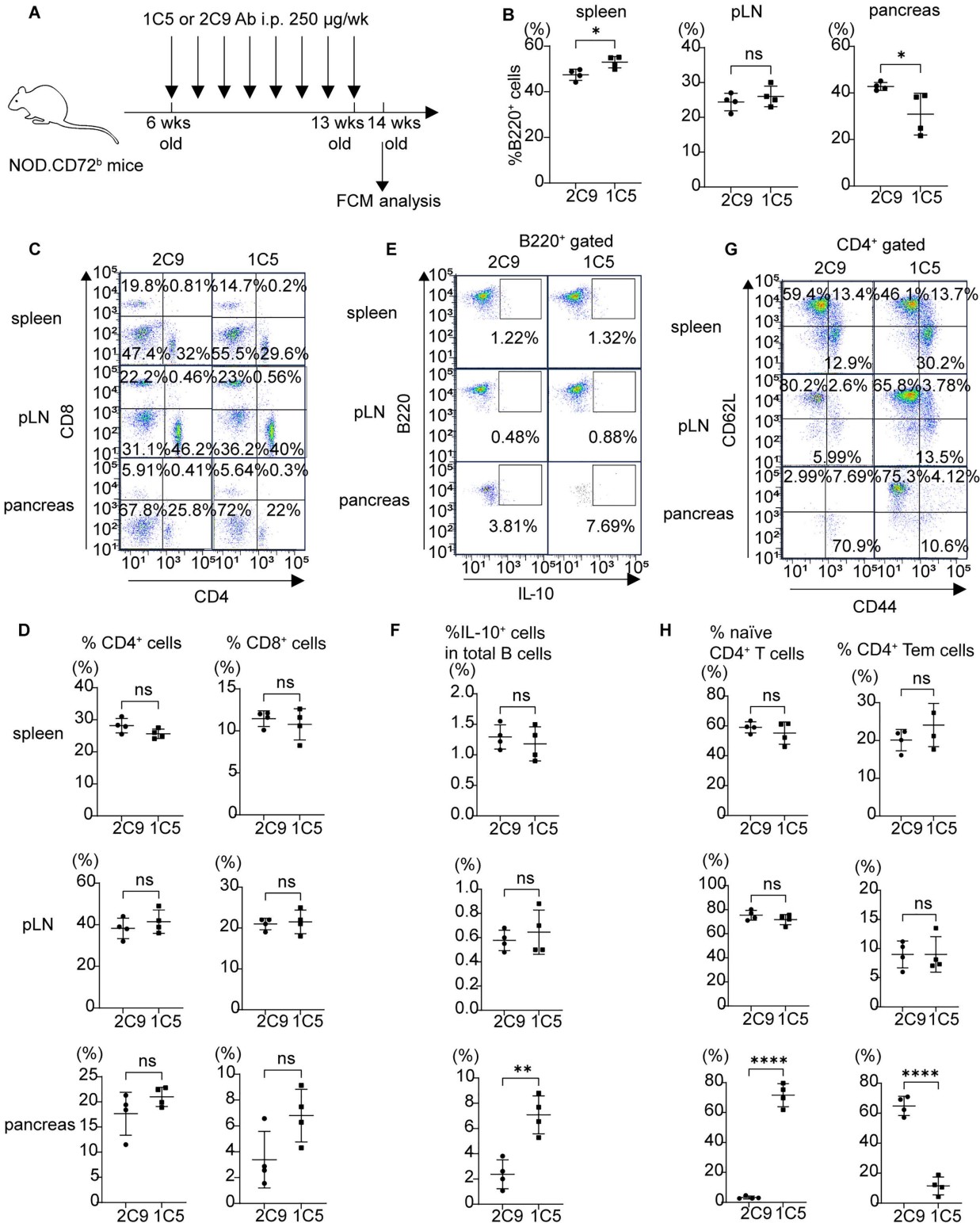

**Fig 8. Expansion of Breg cells and reduction of T cell activation in NOD pancreas by the in vivo treatment with 1C5 require inhibition of the ligand binding of CD22. (A)** Schematic representation of the experimental procedure. Female NOD.CD72b mice were i.p. injected with 250 µg 1C5 or

2C9 weekly from 6 to14 weeks **(B, E)** FCM analysis for B220$^+$ B cells **(B)**, CD4$^+$CD8$^-$, CD4$^-$CD8$^+$ T cells **(C, D)**, IL-10$^+$ B220$^+$ B cells **(E, F)**, CD62L$^+$C-D44$^{lo}$ naïve CD4$^+$ T cells, and CD62L$^-$CD44$^{hi}$ CD4$^+$ Tem cells **(G, H)** in spleen, pancreatic lymph nodes and pancreas from 1C5 and 2C9-treated 15 weeks old mice. Representative data are shown. Percentages of CD4$^+$, CD8$^+$ T cells **(D)**, IL-10$^+$ B cells **(F)**, CD62L$^+$CD44$^{lo}$ naïve CD4$^+$ T cells, and CD62L$^-$CD44$^{hi}$ CD4$^+$ Tem cells **(H)** are indicated. Mean±SD are shown ($n = 4$) (D, F, **H**). Data were analyzed by unpaired $t$ test. *$p < 0.05$, **$p < 0.01$, ****$p < 0.0001$, ns, not significant. Data are representative of two independent experiments **(A–H)**. Numerical data underlying this figure can be found in S1 Data, sheet "Fig 8".

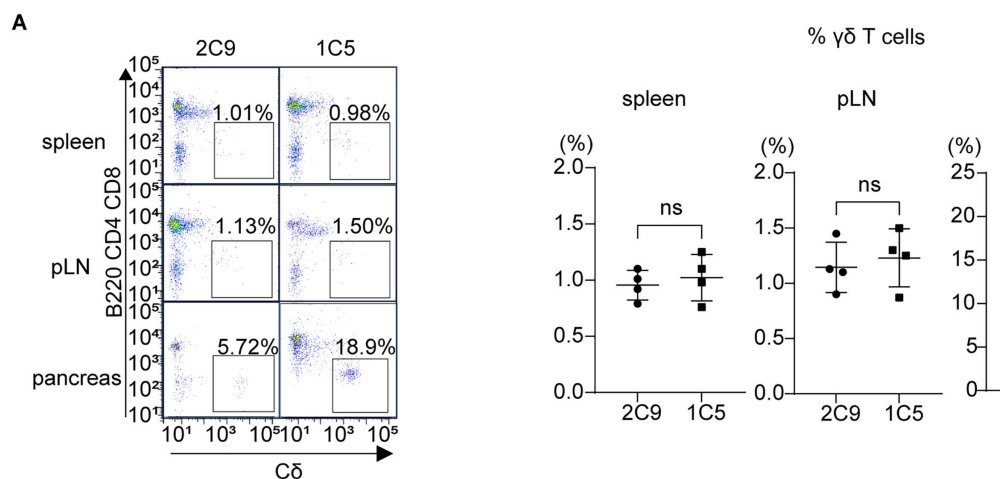

**Fig 9. Expansion of γδ T cells in NOD spleen requires inhibition of the ligand binding of CD22.** Female NOD.CD72$^b$ mice were treated with 1C5 or 2C9 as shown in Fig 8A. Cells from spleen, pancreatic lymph nodes and pancreas are analyzed by FCM. Percentages of B220$^-$CD4$^-$CD8$^-$Cδ$^+$ cells are indicated. Mean±SD are shown ($n = 4$). Data were analyzed by unpaired $t$ test. ***$p < 0.001$, ns, not significant. Data are representative of two independent experiments. Numerical data underlying this figure can be found in S1 Data, sheet "Fig 9".

(InvivoGen). Mice were boosted with the same antigen solution at 2 weeks after the primary immunization. Blood was collected at 2 weeks after primary immunization and 1 week after the boost.

## ELISA

ELISA plates (96-well) were coated with 2 µg/mL OVA in PBS, and blocked with PBS containing 1% BSA. After incubation with serially diluted sera, wells were incubated with alkaline phosphatase-conjugated goat anti-mouse IgG (SBA). After washing, the wells were incubated with the alkaline phosphatase substrate solution (Sigma-Aldrich), and optical density at 405nm was measured by a microplate reader (Molecular Devices).

## Skin transplantation

C57BL/6 mice (12 weeks old) were anesthetized with an isoflurane vaporizer. BALB/c mice (8–12 weeks old) were sacrificed, and the dorsal side of ear skin from BALB/c mice was transplanted to the back of C57BL/6 recipients as previously described [76]. Some recipients were intraperitoneally injected with 1mg/mL CyA (Fujifilm) (20mg per kg body weight) in 0.9% sodium chloride solution every day from the day of transplantation. Graft survival was monitored daily from 7 days after transplantation.

## Assay for alloreactive antibody

Spleen cells were obtained from 8 to 12-week-old BALB/c mice and 1 × 10$^5$ splenic cells in 100 µL PBS containing 1% BSA and 0.1% sodium azide were stained with 1 µL sera from recipient mice for 30 min on ice. Cells were then stained

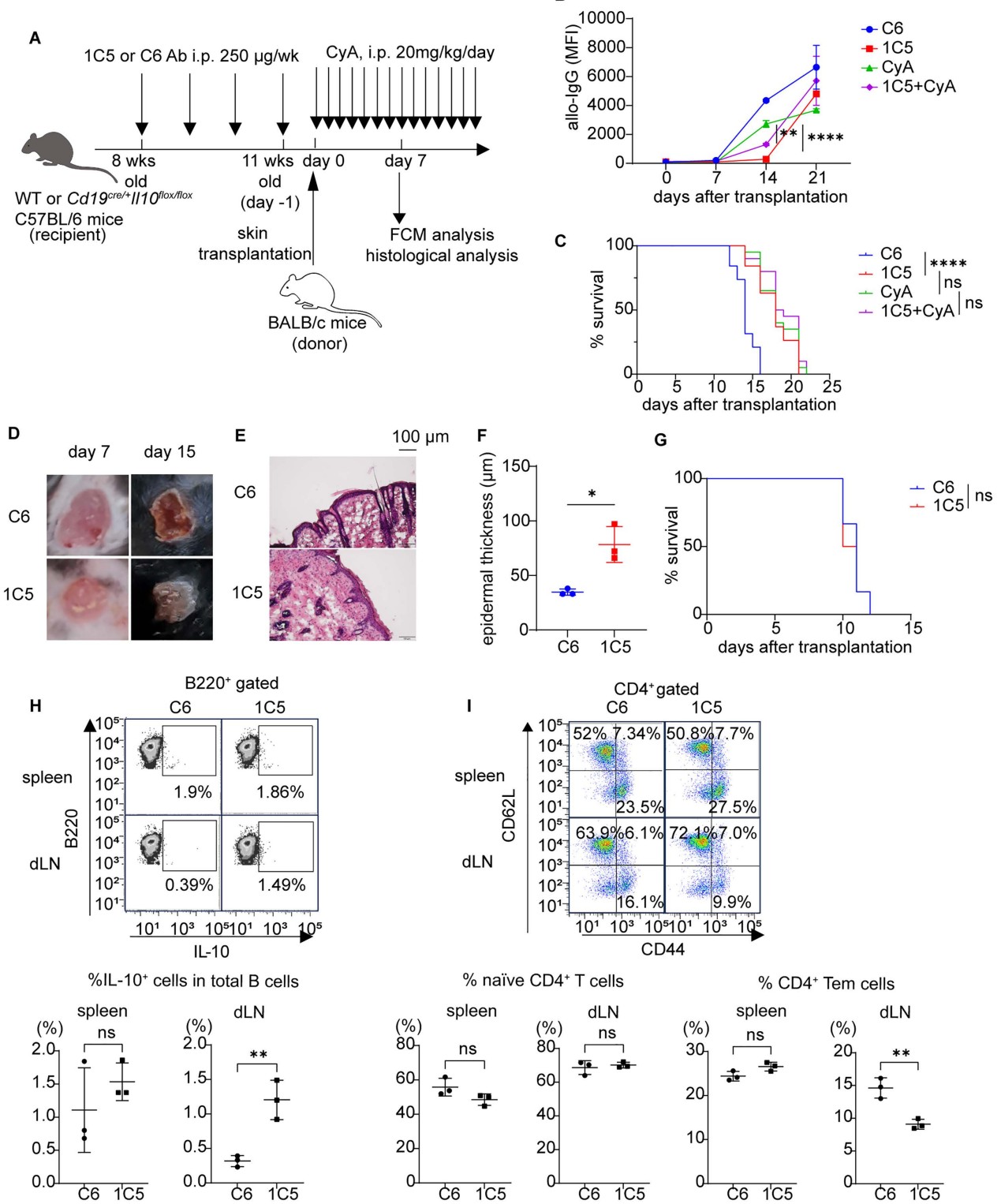

**Fig 10. In vivo treatment with 1C5 prolongs skin allograft survival.** Schematic representation of the experimental procedures. **(B–I)** Wild-type (B–F, H, I) or *Cd19*Cre/+ *Il10*flox/flox **(G)** C57BL/6 mice were i.p. injected 4 times with 250 µg 1C5 or C6 once in a week, followed by transplantation of ear skin from BALB/c mice. CyA (20 mg/kg) was injected every day after transplantation until the graft was rejected **(B, C)**. Alloreactive IgG in sera was measured

by FCM (**B**). Mean ± SD are shown (*n* = 5). Data were analyzed by two-way ANOVA with Šidák multiple comparison test. Representative macroscopic images 7 and 15 days after transplantation (**D**) and graft survival curves (**C, G**) (*n* = 20 for wild-type (**C**) and n = 6 for *Cd19*$^{Cre/+}$ *Il10*$^{flox/flox}$ recipients (**G**)) are shown. The transplanted skin allograft was histologically analyzed 7 days after transplantation. Representative images are shown (**E**). Epidermal thickness was measured (**F**). Mean ± SD is shown (*n* = 3). Data were analyzed by unpaired *t* test. Unstimulated cells from draining lymph nodes 7 days after transplantation were analyzed by FCM for IL-10$^+$ B cells (**H**), CD44$^{lo}$CD62L$^+$ naïve CD4$^+$ T cells, and CD44$^{hi}$CD62L$^-$ CD4$^+$ Tem cells (**I**). Mean ± SD is shown (*n* = 3). Data were analyzed by two-way ANOVA with Šidák multiple comparison test. *$p < 0.05$, **$p < 0.01$, ****$p < 0.0001$, ns, not significant. Data are representative of two (B, E, and F) or three (H and I) independent experiments. Numerical data underlying this figure can be found in S1 Data, sheet "Fig 10".

## Histological analysis

Mice were sacrificed, and the pancreas and skin were isolated. The tissue was fixed with 10% formalin and dehydrated with 70% ethanol. Samples were mounted with Tissue-Tek Optimal Cutting Temperature compound (SAKURA, 4583) and sectioned at 10 μm of thickness. Sections were stained with hematoxylin (Fujifilm) and eosin (Muto Pure Chemical), and microscopically evaluated for the presence of mononuclear infiltrates. Insulitis was scored as described previously [47]: 0, normal islet; 1, peri-insulitis or infiltration of less than 25% of the islet surface area; 2, infiltration of 25%−50% of the islet surface area; infiltration of more than 50% of the islet surface area; 4, infiltration of 100% of the islet surface area.

## Statistical analysis

Data were analyzed using unpaired *t* tests, paired *t* test where applicable, and Kaplan-Meier survival curves were analyzed by log-rank test. For multiple comparisons, one-way ANOVA with Tukey's multiple comparison, and two-way ANOVA with Šidák or Dunnett's multiple comparison were used. The number of mice per group and P values were described in each figure legend. Prism 10 (GraphPad software) was used for the statistical analyses. *P* < 0.05 was considered statistically significant.

## Supporting information

**S1 Fig. The anti-CD22 antibody 1C5 inhibits ligand binding of CD22.** (**A**) Binding of antibodies to CD22. The binding of the indicated antibodies to purified spleen B cells from 10-week-old *Cd22*$^{+/+}$ and *Cd22*$^{-/-}$ C57BL/6 mice were analyzed by FCM. (**B, C**) Inhibition of the ligand binding of CD22. A schematic representation of the experimental procedure is shown (B). The binding of mCD22-Fc to cellular ligands on purified spleen B cells from 10-week-old C57BL/6 mice in the presence or absence of the synthetic sialoside GSC718 or the indicated anti-CD22 antibodies was analyzed by FCM (C). Data are representative of two independent experiments (A–C).
(TIF)

**S2 Fig. 1C5 modulates B cell activation in vitro by inhibiting ligand binding of CD22.** Spleen B cells from 8 to 14 weeks old *Cd22*$^{+/+}$ (**A–F**) or *Cd22*$^{-/-}$ (A, B) mice were cultured as described in the legend to Fig 1. Cells were analyzed by FCM. Representative data are shown (A, C, E). Percentages of alive cells are indicated. Percentages of alive cells and Mean ± SD are shown (*n* = 3) (B, D, F). Data were analyzed by two-way ANOVA with Šidák (B, D) or Dunnett's (F) multiple comparison test. *$p < 0.05$, **$p < 0.01$, ***$p < 0.001$, ****$p < 0.0001$, ns, not significant. Data are representative of two (A–D) or three (E and F) independent experiments. Numerical data underlying this figure can be found in S1 Data, sheet "S2 Fig".
(TIF)

**S3 Fig. In vivo treatment with anti-CD22 antibodies reduces the surface expression of CD22 and MZ B cells but not the total B cell number.** (**A**) Percentages of B220+ cells in spleen cells from 8 to 14 weeks old C57BL/6 mice injected 4 times with 250 μg of the indicated anti-CD22 antibodies or the control antibody C6 every week were analyzed by FCM. Mean ± SD ($n = 3$) is shown. Data were analyzed by one-way ANOVA followed by Tukey's multiple comparison test. (**B, C**) FCM analysis for CD21 and CD23 in spleen B cells from 8 to 14 weeks old female untreated *Cd22+/+* and *Cd22−/−* C57BL/6 mice or *Cd22+/+* C57BL/6 mice injected 4 times with 250 μg of the indicated anti-CD22 antibodies or the control antibody C6 every week. Representative data are shown. Percentages of CD21hiCD23lo MZ B cells are indicated. Mean ± SD ($n = 3$) is shown. Data were analyzed by unpaired *t* test (B) and one-way ANOVA followed by Tukey's multiple comparison test (C). (**D, E**) FCM analysis for the expression of CD22 in spleen B cells from 8 to 14 weeks old female C57BL/6 mice after treatment with the indicated anti-CD22 antibodies in vitro (D) or in vivo (E). Spleen B cells were cultured in the presence or absence of 10 μg/mL of anti-CD22 antibodies for 24 h (D). Spleen cells were obtained from untreated mice or mice sacrificed 1 day or 5 days after injection of 250 μg anti-CD22 antibodies (E). Cells were reacted with the same antibody used for in vitro (D) or in vivo (E) treatment, and the amount of the total anti-CD22 antibody was analyzed by FCM. Representative data and Mean ± SD ($n = 3$) are shown. Data were analyzed by two-way ANOVA with Šidák multiple comparison test (D) or Tukey multiple comparison test (E). *$p < 0.05$, ***$p < 0.001$, ****$p < 0.0001$, ns, not significant. Data are representative of two (B) or four (A, C-E) independent experiments. Numerical data underlying this figure can be found in S1 Data, sheet "S3 Fig".
(TIF)

**S4 Fig. In vivo treatment with 1C5 expands IL-10+ FO B cells.** Spleen cells from *Cd22+/+* (**A**, **B**) and *Cd22−/−* (**A**) *Il10Venus* reporter mice either untreated or treated with the anti-CD22 antibodies 1C5 (**B**, **C**), F239, 1D7, 1D9, or 2C9 (**B**) or the isotype-matched control antibody C6 (**B**, **C**) are stimulated and analyzed as described in the legend to Fig 3. Total number of IL-10+ spleen B cells are calculated by multiplying the number of spleen cells, the percentage of each fraction and the percentage of IL-10+ cells in each fraction. Percentages of IL-10+ B cells in indicated subsets and Mean ± SD are shown ($n = 3$) Data were analyzed by unpaired *t* test (A, C) or one-way ANOVA with Tukey's multiple comparisons (B). ***$p < 0.001$, ****$p < 0.0001$, ns, not significant. Data are representative of two independent experiments (A–C). Numerical data underlying this figure can be found in S1 Data, sheet "S4 Fig".
(TIF)

**S5 Fig. Expression of surface markers is not affected by treatment with PMA/ionomycin.** Spleen cells from 8 to 14 weeks old C57BL/6 mice were cultured with or without 10 ng/mL PMA and 1 μg/mL ionomycin for 5 hours. Expression of the indicated surface markers in B220+-gated (**A–E**) or ungated (**F**) cells was analyzed by FCM. Representative data are shown. Percentages of CD21hiCD23lo, CD21loCD23hi (A), B220+ CD43+ (B), B220+CD1dhi (C), B220+CD5+ (D), B220−/lo CD138+ cells and MFI of IgM and IgD (E) are indicated. Mean ± SD ($n = 3$) is shown. ns, not significant. Data were analyzed by unpaired *t* test. Data are representative of two independent experiments (A–F). Numerical data underlying this figure can be found in S1 Data, sheet "S5 Fig".
(TIF)

**S6 Fig. Expression of CD22 and CD5 in various B cell subsets from *Cd22+/+* and *Cd22−/−* mice.** CD21+CD23+ FO B cells, CD21hiCD23+ marginal zone precursors (MZP), CD21hiCD23− MZ B cells, and CD43+CD23− B-1 cells from *Cd22+/+* (**A**) and *Cd22−/−* (**B**) C57BL/6 spleen were analyzed for expression of CD5 and CD22 by flow cytometry. Data are representative of two independent experiments (A and B).
(TIF)

**S7 Fig. CD22 deficiency expands Tim-1+ IL-10+ B cells.** Spleen B cells from *Cd22+/+* and *Cd22−/−* *Il-10venus/venus* mice were analyzed for expression of Tim-1, CD5 and IL-10 by flow cytometry. Mean ± SD ($n = 3$) is shown. Data were analyzed by

two-way ANOVA with Šidák multiple comparison test. **p < 0.01, ****p < 0.0001, ns, not significant. Data are representative of three independent experiments (A–C). Numerical data underlying this figure can be found in S1 Data, sheet "S7 Fig".
(TIF)

**S8 Fig. Expansion of PBs and PCs by CD22 inhibition.** Spleen cells from 8 to 14 weeks old female $Cd22^{+/+}$ (**A** and **B**) or $Cd22^{-/-}$ (A) C57BL/6 mice either untreated (A) or injected twice (8 days and 1 day before analysis) with 250 μg of 1C5 or C6 (B) were cultured with or without 10 ng/mL PMA and 1 μg/mL ionomycin for 5 hours. Percentage of $B220^{-/lo}CD138^+$ PBs + PCs are indicated. Mean ± SD (n = 3) is shown. Data were analyzed by unpaired t test. *p < 0.05, **p < 0.01. Data are representative of two independent experiments (A and B). Numerical data underlying this figure can be found in S1 Data, sheet "S8 Fig".
(TIF)

**S9 Fig. Analysis of immune cells in NOD.CD72$^b$ mice treated with 1C5 or 2C9.** Female NOD.CD72$^b$ mice were i.p. injected with 250 μg 1C5 (**A–E**), C6 (**A–D**), or 2C9 (**E**) anti-CD22 that does not inhibit ligand binding, every week from 6 to 13 weeks of age and cells from the spleen, pancreatic lymph nodes, and pancreas are obtained at 14 weeks of age (Fig 6A). Number of nucleated cells (A and E). Mean ± SD (n = 4) is shown. FCM analysis for $CD4^+Foxp3^+$ Treg cells (B) and $CD11b^+CD11c^+$ DCs (C). Mean ± SD (n = 3) is shown. Expression of Vγ1, Vγ4, Vγ5, Vγ6, Vγ7 in γδ T cells in pancreas from 1C5-treated 15-week-old mice (D). Representative data are shown. Percentage of $Vγ1^+$, $Vγ4^+$, $Vγ6^+$, $Vγ7^+$ γδ T cells are indicated. Data were analyzed by paired t test (A) or unpaired t test (B, C, E). *p < 0.05, ns, not significant. Data are representative of two independent experiments (A–E). Numerical data underlying this figure can be found in S1 Data, sheet "S9 Fig".
(TIF)

**S10 Fig. Cell numbers of various splenic immune cells from NOD.CD72$^b$ mice treated with 1C5 or C6.** Cell numbers of the indicated subsets were calculated by multiplying total cell number and the percentage of each subset shown in Fig 6. Mean ± SD (n = 3 or 4) is shown. Data were analyzed by unpaired t test. ns, not significant. Data are representative of two independent experiments. Numerical data underlying this figure can be found in S1 Data, sheet "S10 Fig".
(TIF)

**S11 Fig. Cell numbers of various immune cells from pancreatic lymph nodes and pancreas from NOD.CD72$^b$ mice treated with 1C5 or C6.** Cell numbers of the indicated subsets from pancreatic lymph nodes (**A**) or pancreas (**B**) were calculated by multiplying total cell number and the percentage of each subset shown in Fig 6. Mean ± SD (n = 3 or 4) is shown. Data were analyzed by unpaired t test (A) or paired t test (B). *p < 0.05, ns, not significant. Data are representative of two independent experiments (A and B). Numerical data underlying this figure can be found in S1 Data, sheet "S11 Fig".
(TIF)

**S12 Fig. Detection of IL10 production in B cells using *Il10*Venus reporter and anti-IL-10 antibody.** Spleen cells from 8 to 14 weeks old C57BL/6 mice or $Il10^{Venus}$ reporter mice stimulated with or without 10 ng/mL PMA and 1 μg/mL ionomycin for 5 hours were stained with or without anti-IL-10 antibody and analyzed by FCM. Mean ± SD (n = 3) is shown. Data were analyzed by one-way ANOVA followed by Tukey's multiple comparisons test. *p < 0.01, ***p < 0.001, ****p < 0.0001, ns, not significant. Data are representative of two independent experiments. Numerical data underlying this figure can be found in S1 Data, sheet "S12 Fig".
(TIF)

**S13 Fig. Cytokine secretion from T cells in NOD. CD72$^b$ mice treated with 1C5.** Lymph node cells from a 8 week-old C57BL/6 mouse (**A**, **B**) and cells obtained from spleen, pancreas lymph node (pLN) and pancreas were stimulated with 10 ng/mL PMA and 1 μg/mL ionomycin for 5 hours, and production of IFN-γ (**C**, **E**) and TNF-α (**D**, **F**) in $CD4^+$ (**C**, **D**) and

CD8$^+$ (**E**, **F**) T cells were analyzed by FCM. Alternatively, cells were not stained with antibodies to cytokines (A and B). Mean ± SD ($n = 3$) is shown. Data was analyzed by unpaired $t$ test. *$p < 0.05$, ns, not significant. Data are representative of two independent experiments (A–F). Numerical data underlying this figure can be found in S1 Data, sheet "S13 Fig".
(TIF)

**S14 Fig. Analysis of immune cells in in vivo 1C5-treated recipient mice transplanted with skin allograft.** C57BL/6 mice were i.p. injected with 250 μg 1C5 or C6 every week from 8 to 11 weeks of age, and transplanted with ear skin from 11 weeks old BALB/c mice (Fig 8A). Cells were obtained from draining lymph nodes (axillary lymph nodes) and spleen 7 days after transplantation. (**A**) Nucleated cell numbers in spleen and draining lymph nodes (**B and C**) FCM analysis of untreated cells for B220$^+$ B cells, CD3$^+$CD4$^+$ T cells, CD3$^+$CD8$^+$ T cells, CD11b$^+$CD11c$^+$ DCs and CD4$^+$Foxp3$^+$ Treg cells from spleen (B) and draining lymph nodes (C). Mean ± SD ($n = 3$) is shown. (**D**, **E**) Cells were stimulated with 10 ng/mL PMA and 1 μg/mL ionomycin for 5 hours, and analyzed for IFN-γ and TNF-α in CD4$^+$ and CD8$^+$ T cells from spleen (D) and lymph nodes (E). Mean ± SD ($n = 3$) is shown. Data was analyzed by unpaired $t$ test. ns, not significant. Data are representative of three independent experiments (A–E). Numerical data underlying this figure can be found in S1 Data, sheet "S14 Fig".
(TIF)

**S15 Fig. Numbers of various immune cells in in vivo 1C5-treated recipient mice transplanted with skin allograft.** Cell numbers of the indicated subsets from spleen (**A**) or draining lymph nodes (**B**) were calculated by multiplying total cell number and the percentage of each subset shown in Figs 10 and S12. Mean ± SD ($n = 3$ or 4) is shown. Data were analyzed by unpaired $t$ test. *$p < 0.05$, **$p < 0.01$, ns, not significant. Data are representative of three independent experiments (A and B). Numerical data underlying this figure can be found in S1 Data, sheet "S15 Fig".
(TIF)

**S16 Fig. In vivo treatment with 1C5 does not alter antibody production to a T cell-dependent antigen.** Eight weeks old C57BL/6 mice were 4 times i.p. injected with 250 μg 1C5 or C6 every week. Mice were immunized with 10 μg OVA in 100 μL PBS with 100 μL alhydrogel on the day of the final injection of the antibody, and boosted with the same antigen (**A**). Sera were collected at the indicated time and the antibody titer was measured by ELISA (**B**). Mean ± SD ($n = 4$) is shown. Data was analyzed by two-way ANOVA with Šidák multiple comparison test. ns, not significant. Data are representative of two independent experiments (A and B). Numerical data underlying this figure can be found in S1 Data, sheet "S16 Fig".
(TIF)

**S1 Table. List of antibodies used for FCM in this study.**
(PDF)

**S1 Data. Numerical values of data in Figs 1–10, S2–S5, and S7–S16.**
(XLSX)

## Acknowledgments

We thank Drs. K. Rajewsky (Max Delbruck Center) for *Cd19*$^{cre/cre}$ mice, A. Roers (Heidelberg University) for *Il10*$^{flox/flox}$ mice, K. Takeda (Osaka University) for *Il10*$^{Venus}$ mice, and late Dr. T. F. Tedder for *Cd22*$^{-/-}$ mice. We also thank Drs. H. Ishida (Gifu University) and T. Azuma (Tokyo University of Science) for reagents, Shinji Kunitake and Hewessa Gamage Nadeesha Gayathree Gamage (Institute of Science Tokyo) for technical help. Dr. Masatake Asano passed away before the submission of the final version of this manuscript. Takeshi Tsubata accepts responsibility for the integrity and validity of the data collected and analyzed.

## Author contributions

**Conceptualization:** Takeshi Tsubata.

**Funding acquisition:** Takeshi Tsubata.

**Investigation:** Wang Long, Ayaka Endo, Hashadi Nadeesha Walakulu Gamage, Tianyi Yang.

**Methodology:** Toru Suzuki, Takeshi Nitta, Hiromu Takematsu, Kenya Honda.

**Supervision:** Masatake Asano.

**Writing – original draft:** Wang Long, Takeshi Tsubata.

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
