## [Editor Report · Decision Letter 0]

16 May 2025

Dear Takeshi,

I hope you are doing great. Thank you for submitting your manuscript entitled "Inhibition of endogenous ligand binding of CD22 ameliorates type 1 diabetes and graft rejection by expanding regulatory B cells" for consideration as a Research Article by PLOS Biology.

Your manuscript has now been evaluated by the PLOS Biology editorial staff, as well as by an academic editor with relevant expertise, and I am writing to let you know that we would like to send your submission out for external peer review.

Once your full submission is complete, your paper will undergo a series of checks in preparation for peer review. After your manuscript has passed the checks it will be sent out for review. To provide the metadata for your submission, please Login to Editorial Manager (https://www.editorialmanager.com/pbiology) within two working days, i.e. by May 18 2025 11:59PM.

Best wishes,

Melissa

Melissa Vazquez Hernandez, Ph.D.

Associate Editor

PLOS Biology

---

## [Decision Letter · Decision Letter 1]

26 Jun 2025

Dear Takeshi,

Thank you for your patience while your manuscript " Inhibition of endogenous ligand binding of CD22 ameliorates type 1 diabetes and graft rejection by expanding regulatory B cells " was peer-reviewed at PLOS Biology. It has now been evaluated by the PLOS Biology editors, an Academic Editor with relevant expertise, and by two independent reviewers.

In light of the reviews, which you will find at the end of this email, we would like to invite you to revise the work to thoroughly address the reviewers' reports. As you will see below, the reviewers are positive about the relevance of the study, yet some concerns have raised during revision. Reviewer 1 has some concerns regarding replicates, as this are not mentioned in some figures, and request that you show the expression levels of CD22 and if the increase of gamma delta T cells in pancreas in 1C5 treated mice is specific to blocking cis ligand interactions. Reviewer 2 have several technical concerns and also mentions the study is somehow incomplete.The reviewer requires some clarifications which in several cases will require further experiments that could also make the work more complete. We agree with all reviewer concerns and would require some additional experimental revisions to address them, as we consider that this would strengthen the work.

Given the extent of revision needed, we cannot make a decision about publication until we have seen the revised manuscript and your response to the reviewers' comments. Your revised manuscript is likely to be sent for further evaluation by all or a subset of the reviewers.

**IMPORTANT - SUBMITTING YOUR REVISION**

*Re-submission Checklist*

*Published Peer Review*

*PLOS Data Policy*

*Blot and Gel Data Policy*

Sincerely,

Melissa

Melissa Vazquez Hernandez, Ph.D.

Associate Editor

PLOS Biology

REVIEWERS' COMMENTS

Reviewer #1 (Claudia Mauri):

This is an interesting study from Long et al, characterising how anti-CD22 antibodies promote the generation of regulatory B cells. The authors build on the published literature by showing that, in particular, expansion of Bregs in CD22 deficient mice is driven by blocking CD22 interactions with its cis sialic acid ligands. The authors show that treatment of mice with 1C5 blocks cis ligand interactions with CD22 and this promotes Breg generation. Furthermore, the authors show treatment with this antibody delays the onset of diabetes and skin allograft rejection in mouse models.

Major comments

1. A lot of the figures, particularly figures 1-4 have an N number of 3 and it is not stated whether these results are one or more independent experiments. Please indicate this, and if necessary, build the numbers to strengthen the validity of the findings.

2. the authors should include data showing the CD22 expression levels amongst different B cell subsets. In particular how does CD22 expression differ in CD5+ and CD5- follicular B cells. Do CD5+ FO B cells have a lower expression of CD22 for instance, which helps promote their regulatory function?

3. Can the authors explain as to why the proliferation of B cells in figure 1E is so high in the unstimulated B cells treated with 1C5?

4. In figure 7 where the authors look at the gamma delta T cells in NOD.CD72b mice and see an increase in this population in the pancreas of 1C5 treated mice, is this a feature specific to blocking cis ligand interactions? Have the authors examined this with the other CD22 blockign antibodies F239, 1D7 etc. In this model, is this phenotype driven by CD5+ FO B cells?

Minor comments

1. In figure 3, the graphs are not in order i.e. G and F.

2. Can the authors speculate in the discussion as to Bregs promote the expansion of CD39+ gammadeltaT cells?

Reviewer #2:

Regulatory B cells modulate the immune response in a wide variety of disease models. New approaches to induce Bregs in vivo are necessary if we are to capitalize on their therapeutic potential. Loss of CD22, a B cell specific inhibitor of activation, was previously shown to increase CD1d+CD5Hi "B10" cells and the overall frequency and number of IL-10+ B cells. Here, Long et al develop an anti-CD22 mAb that blocks CD22:ligand interactions and increases the frequency of IL-10+ B cells. Treatment of mice with this mAb inhibits the induction of T1D in NOD mice and prolongs skin allograft survival. The data are straightforward, but somewhat incomplete and several technical concerns need to be addressed. Nonetheless, this work provides a new approach towards in vivo expansion of this important regulatory cell population.

Concerns (addressed section by section):

1. In vitro treatment with IC5 inhibits ligand binding: Figs 1 and S1 are convincing.

-However, proliferation was performed at 72h of culture. B cells survive very poorly without stimulation meaning that the (-) control panel in Fig 1F may be misleading. In panels where cells were stimulated, were the viability or final cell number examined? (Viability dyes in flow cytometry assays is not mentioned in Methods). Large differences in cell survival /number could markedly alter interpretation. This applies to all in vitro experiments (e.g. Fig S2D) and later measurement of intracellular cytokines, where eliminating dead cells from the FACs analysis is critical to reduce background.

- n=3 in most panels, a bare minimum that could hide differences where variability is observed.

2. In vivo treatment with IC5 inhibits CD22 activity:

-Even though overall B cell % does not change, individual B cell subsets (e.g. MZ B cells) could still be deleted by anti-CD22 treatment.

-The authors interpret the changes in IgM and MZ B cells as showing that other anti-CD22 mAbs "partially inhibit" CD22 whereas IC5 "strongly inhibits CD22 activity" (lines 178-179). This is an overinterpretation since CD22 plays a complex role with both ligand-dependent and ligand-independent functions. (For example, CD22 deletion inhibits BCR-mediated signals, but augments BCR+CD40-mediated signals on all/many B cells -not just Bregs). Thus, these mAbs alter CD22 function and CD22 expression, but there is no specific way to measure CD22 "activity" per se.

-Anti-CD22 mAbs that do not inhibit ligand interactions reduce IgM levels and MZ B cells, whereas IC5 which blocks ligand binding, has additional effects to reduce IgD expression, augment CD5 expression and augment IL-10 expression. Why interfering with ligand binding has these additional effects is unclear. As such, the lengthy speculation about possible mechanisms on page 9 should be avoided, particularly in Results section.

-Do different subsets of B cells start with different levels of CD22 expression and could this help explain the effect of anti-CD22 mAbs on certain subsets (e.g. MZ B cells, IgM expression, IL-10+ B cells)?

3. In vivo IC5 expands follicular Breg cells. (More precisely, IC5 expands follicular IL-10+ B cells, since at this point in the manuscript, they have not been demonstrated to have Breg activity).

-littermate fluorescence controls for Venus expression should be shown for representative FACs plots (e.g. supplementary data).

-The results n Fig 3B show that in unimmunized mice treated with control mAb C6 almost >5% of total B cells express the IL-10 reporter. This is markedly higher than other published reports of IL-10 expression by total B cells even in immunized mice - even using the same Venus IL-10 reporter (See Matsumoto Ref 36, Fig 1). This is even more true in Fig 3C) where >50% of MZ B cells express IL-10 - rivaling that of plasma cells (This is a log order difference compared to other publications including Matsumoto (Fig S1C) in immunized mice!). In Fig 3C, in naïve mice without any mAb treatment (i.e. no C6) almost 4% of FO B cells express IL-10. Do the authors have any explanation? Does antibody staining corroborate higher levels than other published results?

-Mice were treated weekly, but it is unclear for how many weeks in these experiments. Was this length of treatment required to see increased IL-10+ B cells?

-Above concerns aside, Fig 3C-H clearly show the increase in IL-10 induced by loss of CD22 or IC5 treatment, primarily amongst FO B cells - interesting since these are usually so low and few studies have focused on their Breg activity.

-The basis for this entire manuscript was the observation that CD22-/- mice express increased CD5+CD1Dhi "B10" cells. Does this "B10" population increase after IC5?

-Kuchroo and others have shown the expression of IL-10 and a number of other anti-inflammatory molecules are preferentially expressed on Tim-1+ B cells and Tim-1 regulates their expression. Loss of Tim-1 on B cells results in spontaneous autoimmunity. Do the FO B cells that increase IL-10 expression after IC5 treatment express Tim-1? Or is this a TIM-1-indepedent Breg subset? This is important because if they express Tim-1, they may also express other regulatory molecules in addition to IL-10. Alternatively, anti-CD22 may induce a completely different kind of Breg that does not involve TIM-1 signaling. These data would further provide further insight into Bregs.

-In all cases, cell number as well as % of IL-10+ B cells/subsets should be shown (E.G. see Yanaba et al; Ref #21). This is critical in evaluating the in vivo response to the anti-CD22 mAbs in both naïve mice and in disease models evaluating T and B cell responses at different sites where changes in percentages may not matter if the cell number is markedly altered.

-The data in Figs 4A and B, are superfluous and not very helpful, since B cell subsets are generally defined by combinations of markers and not any single marker. This includes looking at CD5 expression on B1a cells vs. FO cells. Show the subset and CD5 expression. This is important given that CD5 is not usually associated with FO B cells, Tedder's group previously reported that loss of CD22 was associated with an increase in CD5+CD1Dhi B cells. (IC5 may increase IL-10 in more than one B cell subset).

-Plasma cell gating used by the authors (inverted triangle) in CD138 vs B220 staining (Fig S3) is highly unusual. This gating leaves out bone fide splenic plasmablasts/plasma cells that are B220low and CD138high. The authors should re-gate in a typical manner that include the entire upper left quadrant and then determine whether plasma cell IL-10 is altered by IC5.

4. IC5 treatment prevents diabetes:

Results are clear that IC5 inhibits development of insulitis and hyperglycemia in NOD.72b mice.

-In terms of interrupting T1D, the data (Fig 5F-H) are incomplete. What happens after 24 weeks? Is disease "stopped", or is hyperglycemia/insulitis just delayed? Does late treatment still induce increased Bregs and γδ T cells?

-Cataloguing the effects of IC5 is critical to initial understanding its effects. The number of CD4 and CD8 T cells, B cells etc. in islets (but also spleen and pLN), in addition to their frequency, must be shown. Showing an overall reduction in nucleated cells is not sufficient, especially in pancreas where contaminating non-hematopoietic cells are present. For example, a fall in total CD4% in pancreas after treatment must mean a rise in another population. Is the fall in CD4 frequency matched by the rise in CD4-CD8- γδ T cells when cell numbers are examined?

-The findings are somewhat descriptive and would benefit from further mechanistic insight. Directly assessing of the role of IL-10+ B cells is difficult in this model given the NOD genetic background. Nonetheless, it would be nice to try to address whether the Bregs themselves prevent hyperglycemia or induce the γδ T cells that then prevent insulitis. For example: Can B cells be depleted at a time point before vs. after γδ T cells are induced to identify the role of γδ T cells? Can γδ T cells be depleted with a mAb or transferred to young NOD mice? Can their suppressive activity be measured in vitro?

-At 14 weeks, there is a two-fold increase in frequency of pancreatic IL-10+ B cells, but probably also a decrease in their numbers. Yet, we presume that these cells inhibited the overall pancreatic inflammatory infiltrate. It would be very helpful in understanding how these Bregs prevent disease if earlier time points were examined. Do the Bregs get there first and in increased numbers prevent the accumulation of inflammatory cells? While still descriptive, longitudinal analysis would provide insight into how Bregs inhibit T1D.

-Curiously, in the T1D model, IL-10+ B cells are not increased in the spleen by IC5. Yet, in naïve mice, loss of CD22 or IC5 treatment increases IL-10+ B cells in the spleen. This again suggests that there may be earlier changes that help set up the changes observed at 14 weeks.

-Minor point: The frequency of CD4, CD8, and CD4-CD8- cells in a B220-neg gate is not an ideal way to assess T cell subset frequency because CD3 was not examined and non-T cells and even CD45- cells may be included. (It does appear that the CD4-CD8- cells in treated mice are largely γδ T cells in Fig 7, but this gating strategy still does not examine innate cells).

5. IC5 prolongs skin allograft survival.

-IC5 prolongs skin graft survival in this very immunogenic allograft model. However, weeks of therapy were given before transplantation was performed. Is such a lengthy time frame required to increase Bregs? Is it possible that IC5 would be effective even before increased Bregs can be detected?

-B cell expressed IL-10 was found to be necessary for prolongation of allograft survival by IC5, however, only 3 mice were examined. This is an insufficient sample size.

-The authors are appropriately careful in interpretating these data. While the data support the hypothesis that IC5 acts by increasing B cell IL-10, loss of B cell IL-10 expression may result in an exaggerated immune response resistant to IC5 or any other "tolerogenic" therapy. Yet, there is no obvious better approach.

-Again cell/subset numbers in spleen and dLN should be shown alongside the frequency.

- Were γδ T cells increased in dLN of IC5-treated skin graft recipients?

6. it is interesting that IC5 inhibited specific allo-antibody (T-dependent), but not anti-Ova Abs. This raises questions about the effect of timing or release of Ag in the two models.

---

## [Editor Report · Decision Letter 2]

26 Jan 2026

Dear Takeshi,

Thank you for your patience while we considered your revised manuscript " Inhibition of endogenous ligand binding of CD22 ameliorates type 1 diabetes and graft rejection by expanding regulatory B cells " for publication as a Research Article at PLOS Biology. This revised version of your manuscript has been evaluated by the PLOS Biology editors, and the Academic Editor..

Based on our Academic Editor's assessment of your revision, we are likely to accept this manuscript for publication, provided you satisfactorily address the remaining editorial points. Please also make sure to address the following data and other policy-related requests.

1) We routinely suggest changes to titles to ensure maximum accessibility for a broad, non-specialist readership, and to ensure they reflect the contents of the paper. In this case, we would suggest a minor edit to the title, as follows. Please ensure you change both the manuscript file and the online submission system, as they need to match for final acceptance:

"Disrupting CD22-cis-ligand interactions ameliorates type 1 diabetes and graft rejection by expanding regulatory B cells"

2) Please add to your Competing Interests the following statement “TT is a member of PLOS Biology’s Editorial Board. The other authors declare that no competing interests exist."

3) Please note that per journal policy, the model system/species (mice) studied should be clearly stated in the abstract of your manuscript.

4) The Ethics statement needs to be a separate, independent (and the first) subheading in the Material & Methods section. It must include the full name of the IACUC/ethics committee that reviewed and approved the animal care and use, as well as the protocol/permit/project license number. https://journals.plos.org/plosbiology/s/ethical-publishing-practice

Please supply the numerical values either in the a supplementary file or as a permanent DOI’d deposition for the following figures:

Figure 1BDF, 2ABCD, 3ABE-H, 4A-D, 5CDEHI, 6BDFH, 7AB, 8BDFH, 9A, 10BCFGHI, S2BDF, S3A-E, S4ABC, S5A-F, S7ABC, S8AB, S9A-E, S10A, S11AB, S12A, S13C-F, S14AB, S15AB, S16B

6) Please cite the location of the data clearly in all relevant main and supplementary Figure legends, e.g. “The data underlying this Figure can be found in S1 Data” or “The data underlying this Figure can be found in https://doi.org/10.5281/zenodo.XXXXX”

7) For figures containing FACS data (Figures 1ACE, 2ABCD, 3A-D, 4A-D, 6CEG, 7AB, 8CEG, 9A, 10HI, S1AC, S2ACE, S3B-E, S5A-F, S6AB, S7ABC, S8AB, S9D, S12A, S13AB), please provide the FCS files and a picture showing the successive plots and gates that were applied to the FCS files to generate the figure. We ask that you please deposit this data in the FlowRepository (https://flowrepository.org/) and provide the accession number/URL of the deposition in the Data Availability Statement in the online submission form.

8) Supplementary files (e.g., excel). Please ensure that all data files are uploaded as 'Supporting Information' and are invariably referred to (in the manuscript, figure legends, and the Description field when uploading your files) using the following format verbatim: S1 Data, S2 Data, etc. Multiple panels of a single or even several figures can be included as multiple sheets in one excel file that is saved using exactly the following convention: S1_Data.xlsx (using an underscore).

9) Please ensure that your Data Statement in the submission system accurately describes where your data can be found and is in final format, as it will be published as written there

10) Per journal policy, if you have generated any custom code during the course of this investigation, please make it available without restrictions. Please ensure that the code is sufficiently well documented and reusable, and that your Data Statement in the Editorial Manager submission system accurately describes where your code can be found. More information on our Code Policy, what and how to share can be found here: https://journals.plos.org/plosbiology/s/code-availability

We expect to receive your revised manuscript within two weeks.

*Published Peer Review History*

*Press*

Sincerely,

Melissa

Melissa Vazquez Hernandez, Ph.D.

Associate Editor

PLOS Biology

---

## [Editor Report · Decision Letter 3]

16 Feb 2026

Dear Takeshi,

Thank you for the submission of your revised Research Article "Disrupting CD22-cis-ligand interactions ameliorates type 1 diabetes and graft rejection by expanding regulatory B cells" for publication in PLOS Biology. On behalf of my colleagues and the Academic Editor, David Nemazee, I am pleased to say that we can in principle accept your manuscript for publication, provided you address any remaining formatting and reporting issues. These will be detailed in an email you should receive within 2-3 business days from our colleagues in the journal operations team; no action is required from you until then. Please note that we will not be able to formally accept your manuscript and schedule it for publication until you have completed any requested changes.

PRESS

Sincerely,

Melissa

Melissa Vazquez Hernandez, Ph.D., Ph.D.

Associate Editor

PLOS Biology
